# Macrophages disseminate pathogen associated molecular patterns through the direct extracellular release of the soluble content of their phagolysosomes

Catherine J. Greene[1,2], Jenny A. Nguyen[2], Samuel M. Cheung[2], Corey R. Arnold[2], Dale R. Balce[2,3,4],
Ya Ting Wang [3], Adrian Soderholm[2], Neil McKenna[2], Devin Aggarwal [2], Rhiannon I. Campden[1],
Benjamin W. Ewanchuk[1], Herbert W. Virgin [3,4] & Robin M. Yates [1,2,5✉]

Recognition of pathogen-or-damage-associated molecular patterns is critical to inflammation. However, most pathogen-or-damage-associated molecular patterns exist within intact microbes/cells and are typically part of non-diffusible, stable macromolecules that are not optimally immunostimulatory or available for immune detection. Partial digestion of microbes/cells following phagocytosis potentially generates new diffusible pathogen-or-damage-associated molecular patterns, however, our current understanding of phagosomal biology would have these molecules sequestered and destroyed within phagolysosomes. Here, we show the controlled release of partially-digested, soluble material from phagolysosomes of macrophages through transient, iterative fusion-fission events between mature phagolysosomes and the plasma membrane, a process we term eructophagy. Eructophagy is most active in proinflammatory macrophages and further induced by toll like receptor engagement. Eructophagy is mediated by genes encoding proteins required for autophagy and can activate vicinal cells by release of phagolysosomally-processed, partially-digested pathogen associated molecular patterns. We propose that eructophagy allows macrophages to amplify local inflammation through the processing and dissemination of pathogen-or-damage-associated molecular patterns.

[1] Department of Biochemistry and Molecular Biology, Cumming School of Medicine, University of Calgary, Calgary, AB T2N 4N1, Canada. [2] Department of Comparative Biology and Experimental Medicine, Faculty of Veterinary Medicine, University of Calgary, Calgary, AB T2N 4N1, Canada. [3] Department of Pathology and Immunology, Washington University School of Medicine, St. Louis, MO 63701, USA. [4] Vir Biotechnology, San Francisco, CA, USA. [5] Snyder Institute of Chronic Disease, Cumming School of Medicine, University of Calgary, Calgary, AB T2N 4N1, Canada. ✉email: rmyates@ucalgary.ca

Macrophages are local conductors of the innate immune system that help orchestrate the balance between the destructive, pro-inflammatory responses, and the repair-focused, anti-inflammatory responses in tissues. Following engulfment of extracellular cargo by phagocytosis, the nascent phagosome matures through multiple fusion events with endosomes and lysosomes resulting in rapid acidification and acquisition of acid hydrolases within its lumen[1]. The resultant hybrid organelle—the phagolysosome—has the membranous and lumenal characteristics of lysosomes and is considered to be the end-of-the-line for phagocytosed microbes and dead cells as they are progressively degraded into basic building blocks, such as amino acids, monosaccharides, fatty acids and nucleotides which are subsequently recycled[2].

Macroautophagy, herein referred to as autophagy, is a principal recycling mechanism of cells[3]. This coordinated process involves the enclosure of internal cell cargo destined for recycling by the double membranous phagophore or isolation membrane. Upon induction of autophagy the acquisition and subsequent nucleation of the isolation membrane is triggered. Two ubiquitin-like conjugation systems are responsible for the elongation of a phagophore membrane large enough to engulf cytoplasmic components. Similar to the maturation of phagosomes, the autophagosome fuses with lysosomes to form the functionally mature, single-membranous autophagolysosome[4,5].

Pathogen-associated molecular patterns (PAMPs) and damage-associated molecular patterns (DAMPs) are evolutionarily conserved motifs within macromolecules of microbes and host cells, respectively. When released in diffusible forms, PAMPs and DAMPs are powerful pro-inflammatory activators of leucocytes[6,7]. Although direct PAMP and DAMP release from microbes and necrotic cells has been demonstrated, the vast majority of potential PAMPs and DAMPs are inaccessible to immune detection as they are located within intact microbes or cells, and are typically within, or complexed with, stable macromolecular structures. Many of these macromolecular structures are not immunostimulatory nor freely diffusible in their native state[8]. For instance, N-formyl-methionine-containing proteins are most abundant within the cytoplasm of microbes and within intact mitochondria, and are not optimally recognized by host formyl-peptide receptors nor readily diffusible, until they are cleaved into smaller N-formyl-methionine-containing oligopeptides[9,10]. Similarly, unmethylated CpG-containing DNA confined within intact microbes or mitochondria are not freely diffusible, nor recognizable by Toll-like receptor 9 (TLR9), until hydrolyzed into CpG oligodeoxynucleotides by lysosomal DNase II[11]. Complete enzymatic hydrolysis of PAMPs and DAMPs, however, renders them immunologically inert as their basic recognizable molecular patterns are destroyed. Hence partial or incomplete digestion of phagocytosed microbes and dead cells has the ability to generate vast quantities of diffusible, optimally immunostimulatory PAMPs and DAMPs. Under our current understanding of phagosomal biology, any immunostimulatory PAMPs or DAMPs generated through partial degradation of phagocytosed microbes or necrotic cells would be sequestered within phagolysosomes rendering them unavailable for detection by neighbouring or recently recruited leucocytes, and eventually rendered inert following their complete digestion to basic building blocks.

With the increased use of microscopic analysis of phagosomal pH in live macrophages using highly time-resolved ratiometric fluorometry techniques, we and others in the field have frequently noticed transient neutralization of mature phagolysosomes containing a variety of cargoes[12]. Interestingly these events were commonly observed in phagolysosomes in primary macrophages, but not in immortalized 'macrophage-like' cell lines. We and others commonly attributed this transient neutralization to phagolysosomal instability with subsequent repair. While phagolysosomal instability is likely the cause of some of these events, the frequency by which some macrophages underwent these events without consequence to cellular health or initiation of cell death pathways was puzzling. Here we describe a cellular process in macrophages wherein the mature phagolysosome fuses with the cellular membrane, allowing bidirectional exchange of soluble material between mature phagolysosomes and the extracellular space, accounting for the transient neutralization of some mature phagolysosomes. Following a fission event from the cellular membrane, the phagolysosome resumes its function, retaining its mature state. We show that the fusion of the phagolysosome with the cellular membrane is dependent on certain autophagic machinery and several proteins previously implicated in the fusion of autophagosomes with lysosomes, and autophagosomes with the plasma membrane. Since the particulate matter within the phagolysosome is retained during the discharge of its soluble contents, differentiating this from vomocytosis of cryptococci from phagosomes[13,14], we have named this process eructophagy— from eructare (Latin: to belch or discharge), with the suffix -phagy in recognition of the connection with phagocytosis and autophagy. Notably, eructophagy is dramatically increased in pro-inflammatory macrophages but is effectively turned off in interleukin-4 (IL-4)-treated macrophages, resulting in the sequestration and complete digestion of phagocytosed cargo in the phagolysosomes of these repair-focused phagocytes. We show that eructophagy is enhanced upon TLR stimulation both extracellularly and intraphagosomally. Notably, during eructophagy, partially digested, phagolysosomal content is released extracellularly, thus making newly generated PAMPs and DAMPs available that were able to activate vicinal leucocytes. Together, these data present a previously uncharacterized mechanism by which macrophages can manipulate the inflammatory–anti-inflammatory nature of the extracellular milieu in an autophagy gene-dependent manner, through controlling the level of PAMPs and DAMPs within the tissue microenvironment: with IL-4-treated macrophages sequestering and fully-degrading (detoxifying) phagocytosed PAMPs and DAMPs to limit bystander activation; and pro-inflammatory macrophages generating and releasing new PAMPs and DAMPs, through release of partially digested material, to activate recently recruited leucocytes and amplify a local inflammatory response.

## Results

**Transient neutralization of lumenal pH is observed in phagolysosomes of primary macrophages containing a variety of cargoes.** As a part of a study to investigate heterogeneity in phagosomal pH we utilized high-content, live-cell imaging of bone marrow-derived murine macrophages (BMMØs) following phagocytosis of fixed *Staphylococcus aureus*, *Saccharomyces cerevisiae* and *Vibrio parahaemolyticus* labelled with the fluorescent pH indicators pH rodo-succinimidyl ester (pHrodoSE) and carboxyfluorescein succinimidyl ester (CFSE), and the environmentally stable reference fluorophore Alexa Fluor 647 succinimidyl ester (AF647SE). As expected, phagosomes quickly acidified as they matured into phagolysosomes. However, as previously noted by us and others[12], we occasionally observed sudden and complete neutralization of lumenal pH of individual phagolysosomes, followed by rapid re-acidification within minutes (Fig. 1a, b, Supplementary Fig. 1 and Supplementary Movie 1). These events occurred in a small proportion of phagolysosomes within a population of macrophages; however, some phagolysosomes were observed to undergo multiple neutralization–re-acidification cycles. Together, this series of

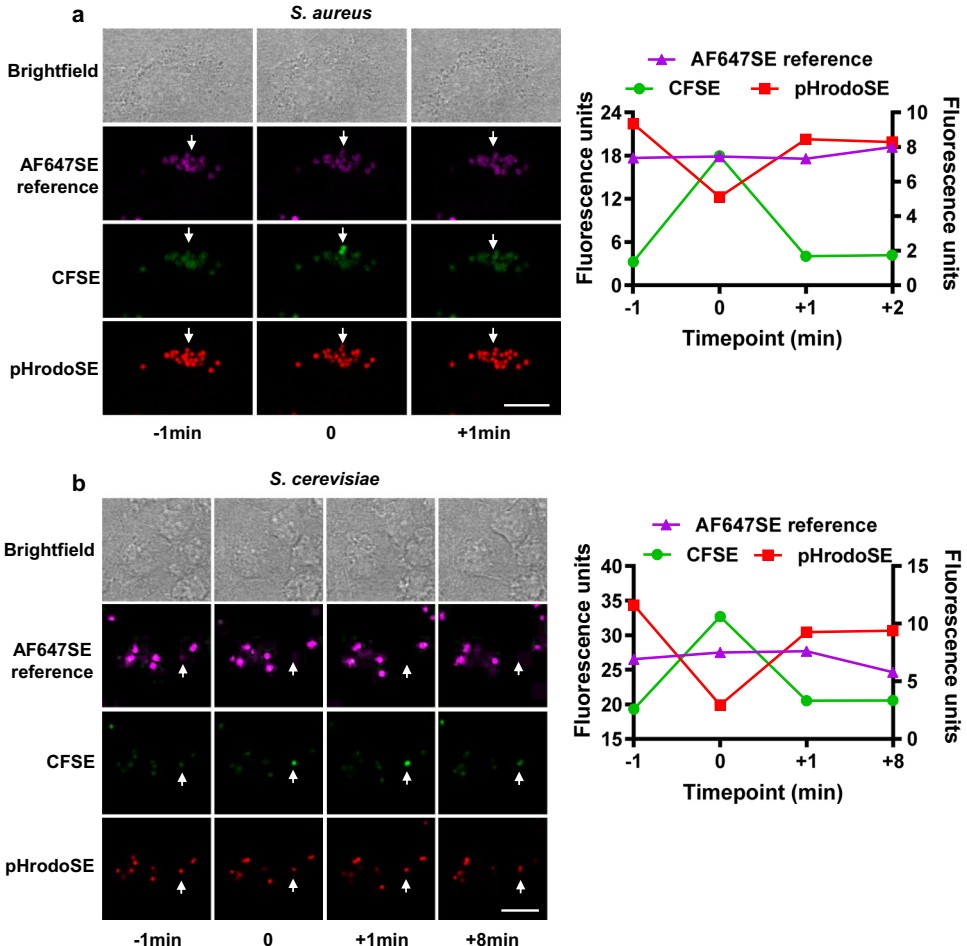

**Fig. 1 Transient neutralization of phagolysosomal lumenal pH after phagocytosis of bacteria and yeast.** Representative sequential images and corresponding fluorescent trace diagrams of BMMØs following the phagocytosis of paraformaldehyde-fixed **a** *Staphylococcus aureus*, **b** *S. cerevisiae* labelled with the pH-sensitive fluors CFSE (green) and pHrodoSE (red) and the reference fluorophore AF647SE (magenta). Transient neutralization of phagolysosomes was demonstrated by concomitant increase in CFSE fluorescence (green) and loss of pHrodoSE fluorescence (red), followed by re-acidification of the phagolysosomal lumen. CFSE and pHRodoSE are plotted on the right axis. AF647SE reference is plotted on the left axis. Images were captured using a Leica SP5 confocal microscope between 45 min and 3 h post-phagocytosis at 37 °C. Time 0 represents a neutralization event. Scale bar denotes 5 μm. Source data are provided as a Source Data file.

experiments demonstrates that through discrete events, mature phagolysosomes of primary macrophages undergo brief periods of lumenal neutralization of pH.

**Soluble content is lost from the phagolysosomal lumen during transient neutralization.** To determine whether the transient neutralization of phagolysosomes was associated with loss of lumenal contents, we pulsed lysosomes with the membrane impermeant fluorescent tracer Alexa Fluor 594 hydrazide (AF594 hydrazide) prior to the phagocytosis of 3.0 μm porous experimental particles labelled with the pH indicator CFSE and the reference fluor AF647SE. As expected, phagosomes rapidly acidified, and accumulated high concentrations of AF594 hydrazide from lysosomes over a 2–3 h period[15]. As seen with phagolysosomes containing bacteria and yeast, following their maturation, we again observed the occasional rapid neutralization of the phagolysosomal lumen, followed by its re-acidification. Interestingly, the rapid neutralization (as indicated by a 'flash' of the pH-sensitive fluorophore, CFSE) precisely corresponded with a complete loss of AF594 hydrazide from the phagolysosome without any detectable leakage into the cytosol of the macrophage (Fig. 2a). To further determine whether partially digested

products generated within the phagosome are lost during these events, we covalently coupled the self-quenched fluorogenic protease substrate (DQ Green BSA) to 3.0 μm porous experimental particles along with the pH indicator (pHrodoSE) and a reference fluorophore (AF647SE) (Fig. 2b)[16]. Following phagocytosis of these particles, as phagosomes matured, their lumens acidified, and proteases progressively digested the fluorogenic protease substrate bound to the particle, liberating fluorescent oligopeptides within the lumen of the phagolysosome. As we previously observed, a portion of phagolysosomes underwent brief neutralization of phagolysosomal pH which coincided with a sudden loss of the fluorescent peptide products. Within minutes, after each loss of peptide event, the phagolysosome quickly re-acidified and resumed proteolytic digestion of the particle-bound substrate (Fig. 2c, d and Supplementary Movie 2). While these events were not observed in all phagolysosomes, certain phagolysosomes underwent multiple digestion-release iterations over a period of 2–10 h after phagocytosis (Fig. 2e and Supplementary Fig. 2). Consistent with prior experiments with the fluorescent tracer AF594 hydrazide, following the sudden loss of these fluorescent peptides from the phagolysosome, we did not observe any increase in cytosolic fluorescence, suggesting that the lumenal contents were being released extracellularly (Fig. 2c and

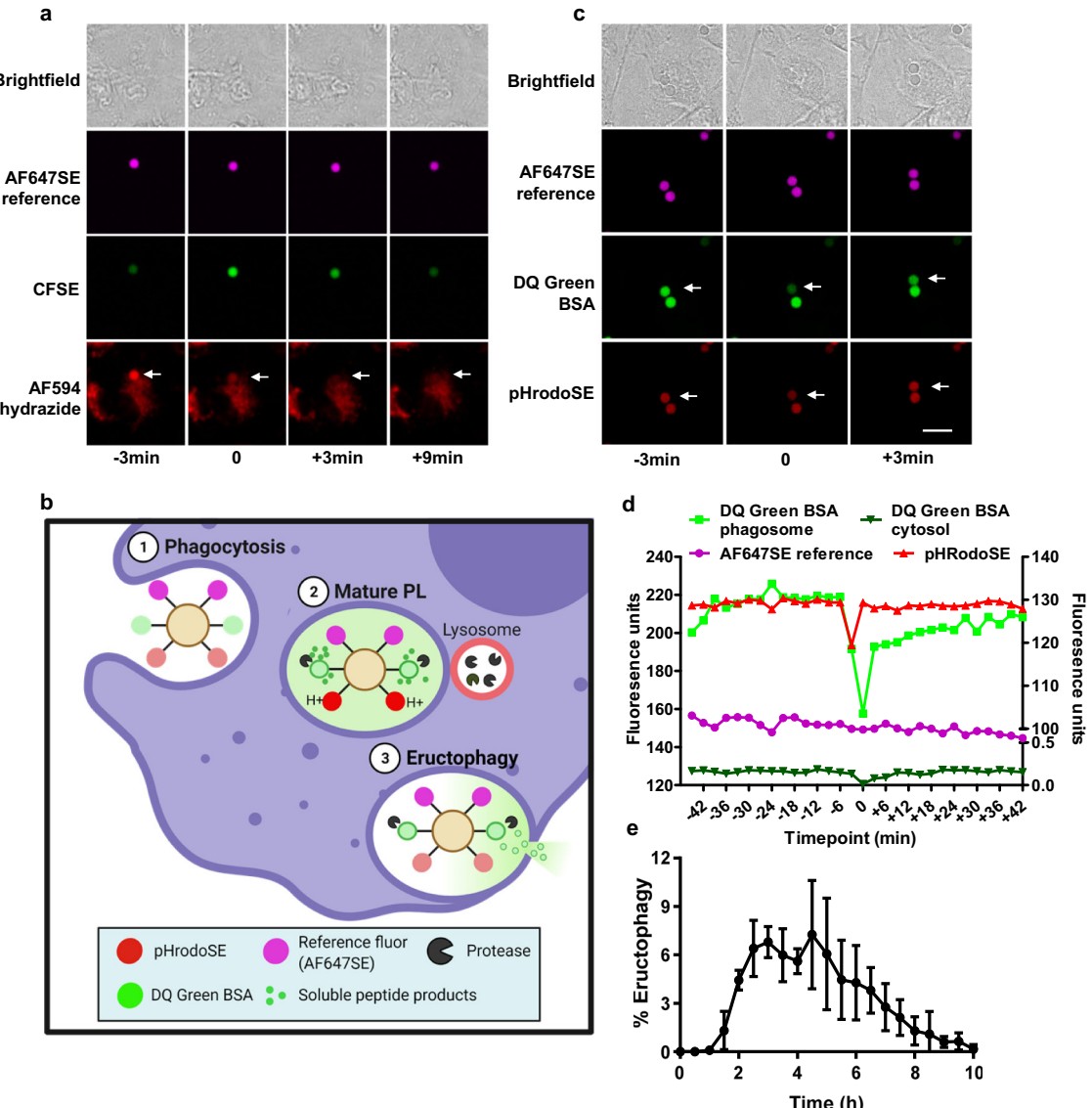

**Fig. 2 Transient communication between the extracellular environment and mature phagolysosomes of pro-inflammatory macrophages. a** BMMØs were pulsed overnight with membrane impermeant fluid-phase tracer AF594 hydrazide (red) and chased to lysosomes over 2 h prior to the phagocytosis of experimental particles labelled with the pH-sensitive fluor CFSE (green) and pH-stable fluor AF647SE (magenta). Sequential images of a phagolysosome containing AF594 hydrazide (by prior fusion with AF594 hydrazide-loaded lysosomes) undergoing transient pH neutralization. AF594 hydrazide fluid-phase fluorophore (red) was lost upon neutralization of the phagolysosome (indicated by the transient increase in CFSE fluorescence on the bead). No increase in cytosolic AF594 fluorescence could be detected following its loss from the phagolysosome indicating extracellular release. **b** Graphical illustration of the experimental particle assay used to measure the loss of partially digested particles from phagolysosomes. The particles bear self-quenched DQ Green BSA, pHrodoSE and the reference fluor AF647SE. **c–e** Phagosomal proteolysis and pH were monitored in BMMØs following phagocytosis of experimental particles bearing the quenched DQ Green BSA substrate (green), the pH indicator pHrodoSE (red) and the reference fluorophore AF647SE (magenta). Eructophagy was demonstrated by the simultaneous loss of soluble peptide products of DQ Green BSA (green) and neutralization of the phagolysosome (red), followed by resumption of proteolysis and re-acidification. **d** DQ Green BSA phagosome and DQ Green cytosol are plotted on the right axis. AF647SE reference and pHRodoSE are plotted on the left axis. **e** Data are presented as the percentage of phagosomes that underwent ≥1 eructophagy event within the corresponding 10 h period post-phagocytosis. Error bars presented as means ± SEM from n = 3 independent experiments. Images were captured using an IN Cell Analyzer 2000 between 90 min and 3 h postphagocytosis at 37 °C. Time 0 represents an eructophagy event. Scale bars denote 10 μm. Source data are provided as a Source Data file.

Supplementary Movie 3). Taken together these data demonstrate that mature phagolysosomes can release soluble products of digestion through discrete events while retaining its particulate cargo—a process we have termed eructophagy.

**Phagolysosomes of macrophages can transiently communicate with the extracellular space.** To further demonstrate that the phagolysosome communicated with the extracellular space as opposed to the cytosol during eructophagy, we employed two approaches. For extracellular communication, we examined whether an extracellular, membrane-impermeable fluorescent substrate could enter the phagolysosome at the same time as the organelle expels the soluble fluorescent products from its lumen. To achieve this, we first loaded lysosomes of BMMØs with fluorescent dextran. We then covalently coupled the enzyme cellulase to experimental particles, in addition to a reference fluor.

Following phagocytosis of these particles by BMMØs, the membrane-impermeant, fluorogenic cellulase substrate (resorufin cellobioside) was added to the assay medium. For the resorufin cellobioside to be hydrolysed (liberating fluorescence), the particle bound cellulase within the phagolysosome must directly communicate with the extracellular space. To determine cytosolic communication, we conducted these assays in GFP$^+$ BMMØs to observe whether the cytosolic GFP would enter phagolysosomes during the loss of fluorescent dextran. Indeed, during the short period following the sudden loss of the fluorescent dextran from the phagolysosome, the quenched extracellular substrate was hydrolyzed within the phagolysosomal lumen by the particle-restricted cellulase enzyme to produce a transient fluorescent signal (Fig. 3a). However, cytosolic GFP did not enter phagolysosomes (Fig. 3a). In contrast, when L-leucyl-L-leucine methyl ester (LLOMe) was used to chemically induce phagolysosomal rupture, cytosolic GFP accumulated in phagolysosomes in tandem with the loss of fluorescent dextran to the cytosol (Fig. 3b). Moreover, only when cells had undergone LLOMe-mediated apoptosis could the particle-restricted cellulase enzyme access its extracellular substrate for hydrolysis (Fig. 3b).

Similar experiments utilizing a variety of particle-bound enzymes and membrane-impermeant substrate combinations, together with fluid-phase fluorescent tracers or pH reporters, further demonstrated a bi-directional exchange of soluble materials between the phagolysosomal lumen and extracellular space during these events (Fig. 3e–g and Supplementary Fig. 3a). To rule out the possibility of the membrane-impermeant extracellular substrate reaching the mature phagolysosomes through gradual pino/endocytic uptake, maturation of the resultant endosomes and fusion with the phagolysosomes, BMMØs containing enzyme-bearing particles were imaged in real time immediately following the addition of the membrane-impermeant substrate to the assay medium. Consistent with direct phagolysosomal communication with the extracellular space, and inconsistent with progressive accumulation of the substrate within the endo/lysosomal network, the hallmark transient fluorescent signals within the phagolysosomes were observed within minutes after substrate addition. (Supplementary Fig. 3b)[17]. The hydrolysis of extracellular substrates by particle-restricted enzymes became our method of choice to detect phagolysosomal communication with the extracellular space, as it produced a readily identifiable fluorescent flash suitable for automated detection, was not substrate-limited, could be used with a variety of experimental particles, and clearly excluded any events associated with phagolysosomal damage (Fig. 3, Supplementary Fig. 4 and Supplementary Movie 4). Together, this series of experiments demonstrate the transient bidirectional exchange of soluble materials between mature phagolysosomes and the extracellular space.

**Eructophagy occurs in primary mouse and human macrophages but not in traditionally immortalized phagocyte-like cell lines**. We next examined whether eructophagy occurs in phagocyte types beyond murine BMMØs. We observed eructophagy in all primary macrophages studied including macrophages isolated from the peritoneum and spleen of mice, and macrophages derived from human blood monocytes (Supplementary Fig. 5a-c). While eructophagy could be detected in primary bone marrow-derived dendritic cells (DC), it was found to occur at a significantly lower frequency compared to the macrophage models tested (Supplementary Fig. 5d). Interestingly, eructophagy could not be observed in traditionally immortalized macrophage-like and DC-like cell lines including RAW264.7, J774, ANA1, PMA-treated THP1 and DC2.4 cell lines but was observed in macrophages derived from conditionally immortalized myeloid precursors expressing ER-HoxB8 (GB8 macrophages) (Supplementary Fig. 5e)[18]. These results suggest that immortalization and continued passage of macrophage-like cells can select against the capacity of cells to execute or properly regulate eructophagy.

**Eructophagy is induced by IFN-γ, inhibited by IL-4 and is negatively regulated by mTOR**. While the rates of eructophagy in resting macrophages were relatively low, with approximately 10% of phagolysosomes undergoing at least one event over a 3 h period, classical activation of macrophages to a pro-inflammatory state with interferon-γ (IFN-γ) was found to significantly increase eructophagy rates (Fig. 4a). In contrast, alternative activation of macrophages to a repair or anti-inflammatory state with IL-4 essentially abolished eructophagy (Fig. 4a). Since IFN-γ is known to downregulate mTORC1 in macrophages[19], we investigated whether mTOR inhibition could further enhance rates of eructophagy. Inhibition of mTOR in IFN-γ-activated macrophages further increased eructophagy rates to a level whereby almost 100% of phagolysosomes underwent at least one eructophagy event over a 5 h period (Fig. 4b). Correspondingly, activation of mTOR with super-physiological levels of essential amino acids abolished eructophagy in IFN-γ-activated BMMØs (Fig. 4c). Together, these data point to a pro-inflammatory role for eructophagy, which is in part controlled by mTOR. This suggests that the level of eructophagy may be higher in tissues when nutritional limitations or immune stimuli responsible for control of tumours or infection are present.

**Eructophagy utilizes autophagic machinery**. The relationship between mTOR and eructophagy led us to explore the potential involvement of autophagy genes in eructophagy. Here, we measured relative rates of eructophagy in BMMØs derived from mice deficient, or conditionally deficient, in key autophagic genes. We found that deficiencies in *Tsc1* (TSC1)[20], *Bcen1* (Beclin 1)[5], *Atg7* (ATG7)[21], *Atg5* (ATG5)[22], *Atg16L1* (ATG16L)[23] and *Pi3kr6* (PI3K)[24] significantly decreased rates of eructophagy (Fig. 5a, b and Supplementary Movie 5). In contrast, deficiencies in genes implicated in LC3-associated phagocytosis (LAP), *Cybb* (NOX2) and *Rubcn* (Rubicon)[5,25] did not impact rates of eructophagy (Fig. 5b), distinguishing eructophagy from the process of LAP. We found that BMMØs deficient in representative autophagy-related genes (Beclin 1, ATG7 and ATG5 and LAP related genes Rubicon) had comparable levels of phagocytic uptake and phagosomal proteolysis (as a functional measure of phagosomal maturation) to WT controls (Supplementary Fig. 6). To investigate the temporal and spatial relationship between autophagy and eructophagy, autophagic signalling was identified by the expression of an LC3-GFP fusion construct in BMMØs following phagocytosis of reporter particles. Interestingly, LC3-GFP recruitment to the phagolysosome began during phagolysosomal communication with the extracellular space and persisted on the phagolysosomal membrane for several minutes after an eructophagy event (Fig. 5c and Supplementary Movie 6)[26]. During eructophagy, the phagolysosomal membrane retained LAMP1, and phagolysosomes were able to fuse with lysosomes immediately following an eructophagic event (Supplementary Fig. 7a, b and Supplementary Movie 7). Together, these data implicate autophagy genes in the initiation and resolution of eructophagy, during which time the phagolysosome can communicate with the extracellular space without losing its mature features or requiring re-maturation.

**Eructophagy is modulated by SNARE complexes associated with degradative autophagy**. The intersection of eructophagy

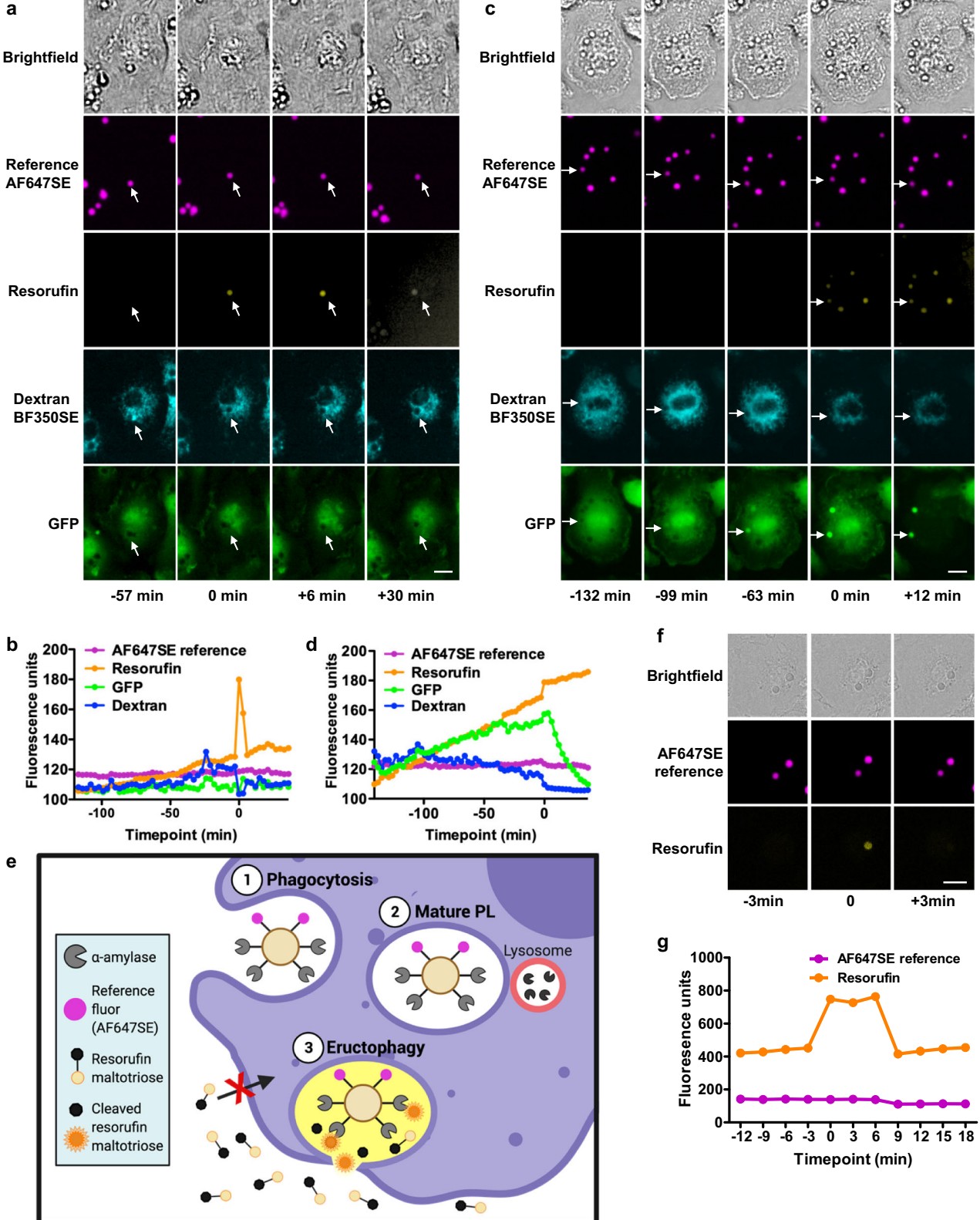

with autophagy and mTOR led us to investigate whether SNARE complexes associated with secretory autophagy play a role in eructophagy[27,28]. Initially, we examined the effects of small hydrophobic molecule inhibitors of SNARE complex zippering on eructophagy. Myricetin is an efficient inhibitor of SNARE zipping through interactions of two specific amino acids common to VAMP2, VAMP8 and Sec22b[29]. Indeed, myricetin significantly

reduced eructophagy events in GB8 macrophages, compared to vehicle or inactive structural analogue (kaempferol) controls (Fig. 6a). Further, ethyl(2-(5-nitrothiophene-2-carboxamido) thiophene-3-carbonyl) carbamate (EACC), a specific Syntaxin 17 inhibitor[30], significantly reduced eructophagic events compared to vehicle controls (Fig. 6a). We next screened GB8 macrophages stably expressing transduced shRNA constructs targeting key

**Fig. 3 Phagolysosomes communicate directly with the extracellular space but not the cytosol during eructophagy.** GFP + BMMØs were pulsed with Bella Fluor (BF) 350SE-dextran (70 kDa) and chased into lysosomes for 2 h. **a**, **b** Phagolysosomes containing cellulase- and AF647SE-coupled reporter particles demonstrate simultaneous loss of dextran from the phagolysosome (cyan) and hydrolysis of the cell impermeant, extracellular substrate (yellow) with no accumulation of GFP⁺ cytosol (green). During eructophagy, the resorufin cellobioside substrate can enter the phagolysosomal lumen where it is cleaved by the particle-restricted cellulase to produce the readily identifiable resorufin fluorescence. **c**, **d** LLOMe (2.5 mM) induced phagosomal rupture, leading to the simultaneous release of dextran (cyan) to the cytosol and accumulation of cytosolic GFP (green) in the phagolysosome, but without the resorufin fluorescence (yellow). **e** Graphical illustration of the experiment depicting particle restricted α-amylase and its cell impermeant substrate resorufin maltotriose. During eructophagy, the resorufin maltotriose substrate can enter the phagolysosomal lumen where it is cleaved by the particle-restricted α-amylase to produce the readily identifiable resorufin fluorescence. **a**, **c**, **f** Representative sequential images acquired with the **a**, **c** IN Cell 2000 Analyzer or **f** Leica SP5 confocal microscope. **b**, **d**, **g** Representative real-time traces of fluorescent intensities from a single phagolysosome. Time 0 represents an eructophagy event. Scale bars denote 10 μm. Source data are provided as a Source Data file.

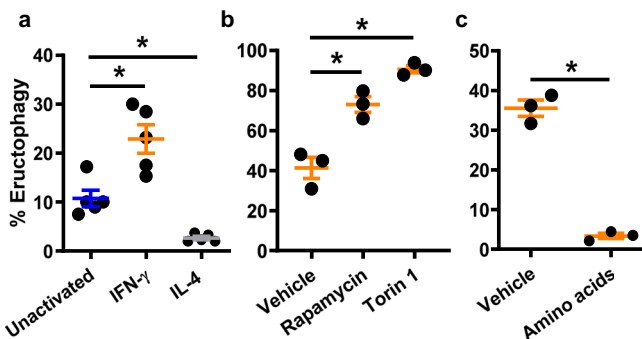

**Fig. 4 Frequency of eructophagy is increased in IFN-γ-activated macrophages and further enhanced with mTOR inhibition.**

**a**–**c** Percentages of phagolysosomes undergoing at least one eructophagic event over 5 h following phagocytosis of particles conjugated to DQ Green BSA, pHrodoSE, AF647SE. Rates of eructophagy are influenced by **a** prior exposure of BMMØs to IFN-γ (100 U/mL, 18 h) or IL-4 (10 ng/mL, 24 h). Rates of eructophagy in IFN-γ-treated BMMØs are influenced by the addition of **b** mTOR inhibitors (rapamycin or Torin 1) or **c** mTOR activators (1× MEM amino acids) 30 min post-phagocytosis. Error bars presented as means ± SEM. **a**, **b** *$P \leq 0.05$ by one-way ANOVA with Bonferroni's post hoc test. **c** $P = 0.001$ Student's two-tailed unpaired $t$-test. **a** $n = 5$ independent experiments. **b**, **c** $n = 3$ independent experiments. Source data are provided as a Source Data file.

membrane fusion and trafficking genes linked to phagosomes, lysosomes, autophagosomes and plasma membrane for their ability to disrupt eructophagy rates (Supplementary Table 1). While many validated knockdowns of targets did not effect eructophagy, knockdown of several targets involved in secretory autophagic vesicle-plasma membrane fusion[31], namely Sec22b, SNAP23, Syntaxin 3 and Syntaxin 4a, had significantly lower rates of eructophagy when compared to a verified non-related gene knockdown controls (Elastase 2) (Fig. 6b and Supplementary Fig. 8). Knockdown of the HOPS complex components Vps33a, Vps39 and Vps41 led to fewer eructophagic events compared to control (Fig. 6c), as did knockdown of vesicle-vesicle fusion proteins Rab7, Syntaxin 17, SNAP29 and VAMP8 (Fig. 6d). These data demonstrate that eructophagy occurs through discrete fusion pathways and implicates fusion and trafficking machinery known to be associated with secretory autophagy.

**Blebbing of the phagolysosomal membrane is associated with eructophagy.** Typical phagosomes lack the machinery to directly fuse with the plasma membrane; however, there are reports demonstrating that phagolysosomes can directly recruit autophagy proteins to its surface that could mediate fusion to the plasma membrane[32,33]. Preceding eructophagy, outpouching or blebbing of the phagolysosomal membrane was routinely observed and was highly predictive of eructophagy events (Fig. 7a

and Supplementary Movie 8). These blebs were frequently decorated with the essential autophagy protein ATG5 by immunofluorescence (Supplementary Fig. 9 and Supplementary Movie 9). We hypothesized that the blebs were an extension of the phagolysosomal membrane through which the phagolysosome interacts with the plasma membrane. We predicted that loss of proteins associated with lysosomal membrane dynamics, such as VAMP8, would result in fewer blebs. Conversely, we predicted that loss of proteins that are known to mediate docking and fusion of lysosomes with the plasma membrane, such as SNAP23, would decrease the resolution of the blebs, resulting in decreased eructophagy, but an increased number of blebs at a given point in time. To test these predictions, the appearance of blebs on mature phagolysosomes containing phagocytosed experimental particles bearing DQ Green BSA were quantified in fixed samples from VAMP8ᵏᵈ, SNAP23ᵏᵈ and control GB8 macrophages. As predicted, phagolysosomes in VAMP8ᵏᵈ macrophages were observed to have a significantly lower incidence of associated blebs, whereas SNAP23ᵏᵈ macrophages were observed to have a significantly more phagosomes with blebs compared to control cells (Fig. 7b). Together these data support the idea that blebbing of the phagolysosomal membrane is a pre-fusion structure, which may be used to connect the phagolysosomal lumen to the plasma membrane to allow for eructophagy.

**Toll-like receptor PAMPs induce eructophagy in a MyD88- and TRIF-dependent manner.** To explore whether TLR activation influenced eructophagy, eructophagy was measured in BMMØs derived from mice deficient in both essential TLR adapter proteins, MyD88 and TRIF following phagocytosis of reporter particles bearing α-amylase from *Bacillus licheniformis*. Indeed, using these reporter particles conjugated to a microbially sourced reporter enzyme, MyD88 and TRIF-deficient BMMØs showed dramatically lower rates of eructophagy (Fig. 8a). Correspondingly, phagolysosomes containing reporter particles devoid of PAMPs displayed similarly low levels of eructophagy in WT, MyD88⁻/⁻ and MyD88/TRIF⁻/⁻ BMMØs (Fig. 8b, c). The adsorption of the TLR9 PAMP, CpG-rich, unmethylated DNA, or the TLR3 PAMP, polyinosinic: polycytidylic acid (PolyI:C), onto the surface of reporter particles dramatically increased eructophagy in a MyD88- and MyD88/TRIF-dependent fashion respectively (Fig. 8b). Eructophagy could also be induced by the addition of the TLR4 PAMP, lipopolysaccharide (LPS), to the assay media following phagocytosis of reporter particles in a MyD88-dependent fashion (Fig. 8c). Together these data demonstrate that the cellular detection of TLR PAMPs induces eructophagy.

**Release of PAMPs by eructophagy can activate vicinal cells.** We hypothesized that eructophagy may function to expose recently recruited leucocytes to PAMPs and DAMPs sequestered in

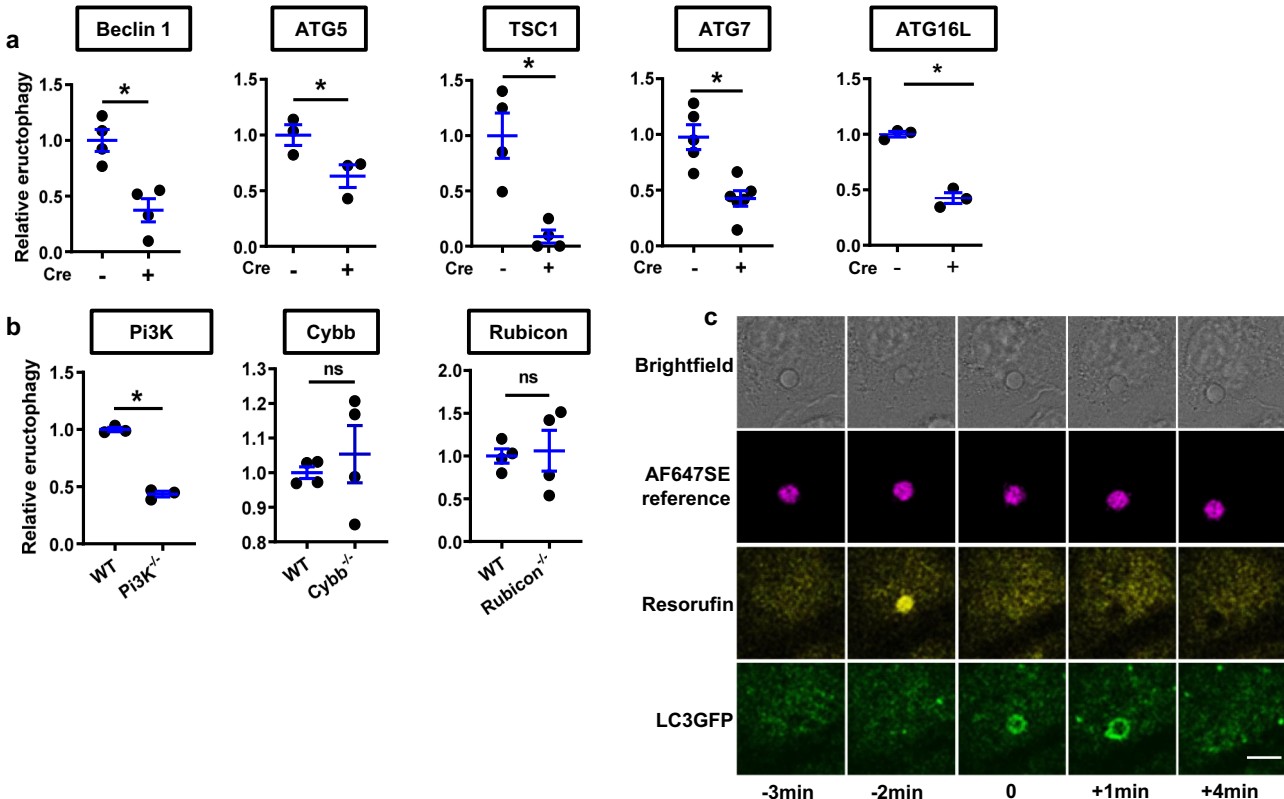

**Fig. 5 Autophagic machinery is required for eructophagy.** Eructophagy was detected by hydrolysis of the membrane impermeant substrate resorufin maltotriose (yellow) by α-amylase conjugated to 3.0 μm reporter particles within phagolysosomes between 1 and 3 h following phagocytosis by BMMØs. Particles are labelled with the reference fluor AF647SE (magenta). **a** Rates of eructophagy in BMMØs conditionally deficient in Beclin 1, ATG5, TSC1, ATG7 and ATG16L (Cre[+]) relative to BMMØs derived from Cre-negative littermates (Cre[−]). **b** Eructophagy is reduced in BMMØs deficient in Pi3Kγ, but not in Cybb or Rubicon relative to BMMØs derived from C57BL6/J mice (wild type; WT). **c** Representative images before, during and after an eructophagy event (yellow) in BMMØs derived from mice expressing LC3-GFP (green). LC3-GFP recruitment to the phagolysosome was observed to be temporally associated with 88% (±3.1%) of all eructophagy events observed in these cells. Error bars presented as means ± SEM. *$P$ = (a) 0.0047, 0.0479, 0.0052, 0.0019 and 0.0004; (b) 0.0001, 0.5513 and 0.8169 by Student's two-tailed unpaired $t$-test. Time 0 represents an eructophagy event. Scale bars denote 10 μm. **a**, **b** $n = 3$, $n = 4$ or $n = 5$ mice per group, as shown by the number of data points on the graph. Source data are provided as a Source Data file.

phagolysosomes of macrophages. To explore this relationship, macrophages phagocytosed experimental particles covalently tethered to CpG-containing plasmid DNA. Diffusion of uncut mitochondrial or microbial DNA is severely limited and it requires hydrolysis into smaller CpG-containing oligodeoxynucleotides (CpG ODN) by lysosomal DNase II before recognition by TLR9[11]. Although TLR9 is traditionally considered an endosomal TLR, many leucocytes express TLR9 on the plasma membrane, physically separated from lysosomal DNase II[34–40]. To determine whether eructophagy may act to release phagolysosomally processed CpG ODN, endotoxin-free, 5.1 kb, CpG-containing plasmid DNA was covalently tethered to experimental particles and given to macrophages deficient in Pi3K and Beclin 1 (two genes that had previously been shown to affect eructophagy), and their corresponding WT controls (Fig. 9a). Following phagocytosis and removal of extracellular particles, the macrophages were co-cultured with reporter cells either expressing human TLR9 (HEK Blue TLR9) or their TLR9-negative parental (HEK Blue Null) cells for 4 h. These reporter cells express the embryonic alkaline phosphatase (SEAP) gene under the IFN-β minimal promoter, which is triggered upon TLR9 activation of NF-κB (Fig. 9a). To determine the amount of cleaved plasmid-derived oligonucleotides released extracellularly by the macrophages, a 66 bp fragment in the CpG-rich region of the plasmid was quantified by qPCR amplification of the supernatants. Indeed, there was significantly more DNA released by WT or Beclin 1 Cre[−/−] than those derived

from Pi3K[−/−] or Beclin 1 Cre[+/−] mice (Fig. 9b, d). Correspondingly, SEAP mRNA expression in the HEK Blue reporter cells was highest in cells co-cultured with WT and Beclin 1 Cre[−/−] BMMØs containing experimental particles with tethered plasmid DNA (Fig. 9c, e). SEAP mRNA expression was virtually undetectable in HEK Blue cells not expressing TLR9 and in cells co-cultured with BMMØs deficient in Pi3K, Beclin 1, or those that had engulfed particles without DNA (Fig. 9c, e). These data demonstrate that microbial CpG DNA is partially digested in the phagolysosome and released to activate TLR9 on vicinal cells. This process is almost completely abolished in eructophagy-impaired cells.

Another PAMP that requires cleavage prior to detection by a pattern recognition receptor is formyl-$N$-methionine-containing peptides. To explore this relationship further, macrophages phagocytosed formyl-$N$-methionine-containing peptides covalently tethered to reporter particles through a cysteine-cathepsin-cleavable region. In this system, to activate the formyl peptide receptor 1 (FPR-1) of neighbouring cells, the tethered formyl-peptide must be first proteolytically liberated from the particle within the macrophage's phagolysosome, and subsequently released to the extracellular space in its partially digested form (Fig. 10a). Lack of digestion of the peptide would result in its sequestration within the phagolysosome, and its complete digestion to individual amino acids and would render the ligand inert. To visualize activation of FPR-1 by liberated and released

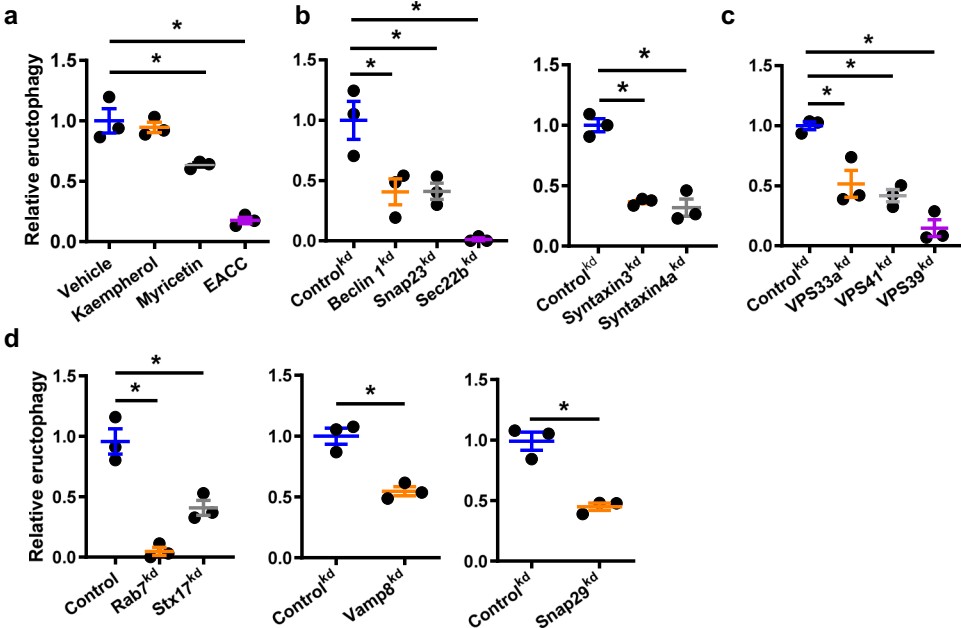

**Fig. 6 Specific SNAREs, Rabs and Syntaxins mediate eructophagy. a–d** Eructophagy was detected by hydrolysis of the membrane impermeant substrate resorufin maltotriose by α-amylase conjugated to 3.0 μm reporter particles within phagolysosomes between 1–3 h following phagocytosis by **a** WT GB8 macrophages or **b–d** GB8 macrophages with the indicated shRNA-mediated knockdown precursor-derived macrophages. Data are presented as relative to control (Elastase 2$^{kd}$). **a** Cells were treated with SNARE inhibitors (10 nM myricetin, 10 nM kaempherol or 100 nM EACC) 1 h post-phagocytosis. **a–d** $n = 3$ per group. **a–d** Error bars presented as means ± SEM. **a–d** *$P \leq 0.05$ by one-way ANOVA with Bonferroni's post hoc test or **d** $P \leq 0.05$ by Student's two-tailed unpaired $t$-test when comparing two groups. $P = 0.0039$ Control$^{kd}$ vs. Vamp8$^{kd}$ and $P = 0.0026$ Control$^{kd}$ vs. Snap29$^{kd}$. Source data are provided as a Source Data file.

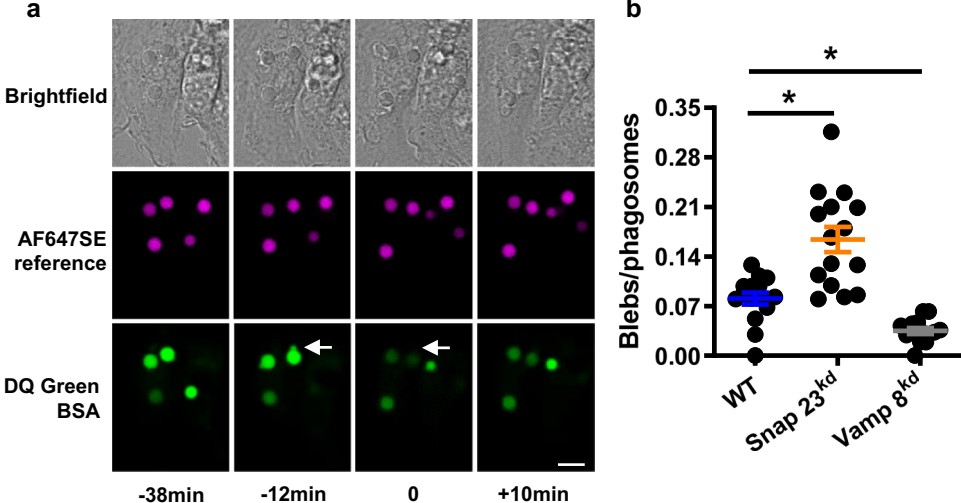

**Fig. 7 Eructophagy is preceded by blebs on the phagolysosomal membrane. a** Representative sequential images of BMMØs following the phagocytosis of experimental particles bearing the quenched DQ Green BSA substrate (green), the pH indicator pHrodoSE (red) and the reference fluorophore AF647SE (magenta). Phagolysosomal morphology can be visualized in the green channel as the soluble DQ Green BSA peptide products are not particle-restricted and hence occupy the entire phagolysosomal lumen. Arrow shows a ~0.5 μm diameter bleb of the phagolysosomal membrane commonly observed before an eructophagic event (observed as the sudden loss of soluble DQ Green BSA peptide products to the extracellular space). Images were captured using a Leica SP5 confocal microscope between 1 and 3 h post-phagocytosis at 37 °C. **b** Quantification of the number of blebs per phagolysosomes following the phagocytosis of experimental particles bearing the self-quenched DQ Red BSA (red) and the reference fluorophore AF647SE (magenta). $N = 15$; 5 fields of view (FOV) over three biologically independent experiments. Error bars presented as means ± SEM. *$P \leq 0.05$ by one-way ANOVA with Bonferroni's post hoc test. Source data are provided as a Source Data file.

formyl-peptides, U937 cells constitutively expressing human FPR-1 (FPR1-U937) were loaded with the calcium indicator Fluo-4 AM and allowed to interact with macrophages that had previously phagocytosed reporter particles with tethered formyl-peptides (Fig. 10a). As anticipated, eructophagic events of

phagolysosomes containing reporter particles with tethered formyl-peptides triggered robust calcium fluxes in vicinal FPR1-U937 cells with a high degree of spatial and temporal association (Fig. 10b and Supplementary Movie 10). To further demonstrate the relationship between eructophagy and PAMP-activation of

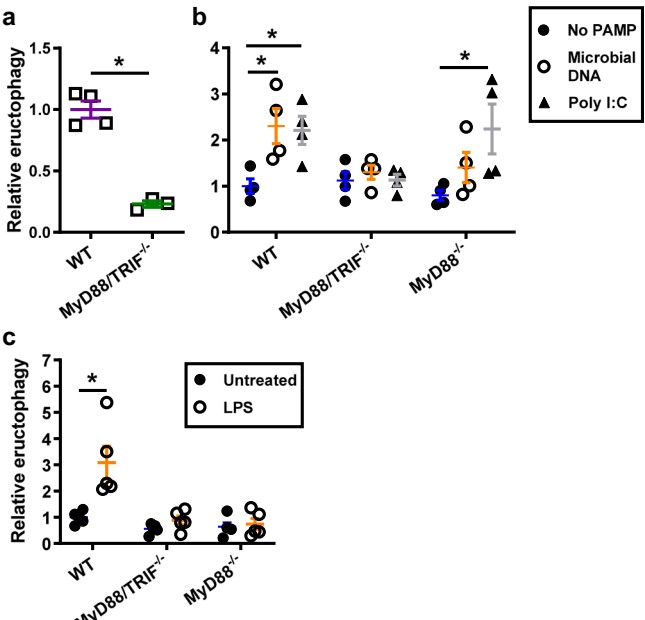

**Fig. 8 Eructophagy in macrophages is induced by TLR activation. a** Rates of eructophagy as detected by extracellular resorufin maltotriose hydrolysis by phagolysosomal α-amylase (sourced from *Bacillus licheniformis*) coupled to reporter particles in BMMØs deficient in MyD88 and TRIF, relative to BMMØs derived from WT mice. **b**, **c** Relative rates of eructophagy as detected by phagolysosomal pH using PAMP-free CFSE-BSA-coupled particles **b** with or without particle-adsorbed TLR ligands (microbial DNA or Poly I:C); or **c** the addition of 10 ng/mL extracellular LPS 30 min post-phagocytosis by BMMØs derived from WT, MyD88/TRIF$^{-/-}$ or MyD88$^{-/-}$ mice. **a**, **b** $n = 4$ biologically independent experiments. **c** $n = 5$ biologically independent experiments. Error bars presented as means ± SEM. by **a** *$P = 0.0003$ Student's two-tailed unpaired *t*-test, or **b**, **c** *$P \leq 0.05$ two-way ANOVA with Bonferroni comparison of means. Source data are provided as a Source Data file.

vicinal cells, BMMØs deficient in MyD88, TSC1 and Beclin 1 (three unrelated genes that had previously been shown to affect eructophagy) were stimulated with extracellular LPS following phagocytosis of reporter particles with tethered formyl-peptides (fMet) or non-formyl peptide control (Met), and then subsequently co-cultured with the FPR1-U937 cells. As expected, higher rates of eructophagy were observed in WT macrophages than in macrophages deficient in MyD88, TSC1 and Beclin 1 (Fig. 10c, e, g). Correspondingly, the frequencies of calcium fluxes in FPR-1-U937 cells, when co-cultured with WT macrophages containing particles with tethered formyl-peptides, were significantly higher than those co-cultured with MyD88-, TSC1- or Beclin 1-deficient macrophages, or macrophages containing particles tethered to non-formylated peptides (Fig. 10d, f, h).

We next set to determine whether the ability of macrophages to activate neighbouring cells through eructophagic release of formyl peptides could be manipulated by IFN-γ and IL-4. Consistent with our data reporting enhanced eructophagy in pro-inflammatory macrophages, IFN-γ activation resulted in higher rates of eructophagy and correspondingly induced higher rates of calcium fluxes in neighbouring FPR1-U937 cells (Fig. 10i, j) and primary murine neutrophils (Supplementary Fig. 10a, b). As expected, IL-4 activation had the opposite effect, showing dramatically reduced rates of eructophagy and activation of neighbouring cells through release of formyl-peptides from their phagolysosomes (Fig. 10i, j and Supplementary Fig. 10a, b). Macrophages containing equivalent reporter particles with non-

formylated peptides did not induce calcium fluxes in co-cultured FPR-1-U937 cells. Hence the association between eructophagy and FRP-1-activation in neighbouring cells is consistent with liberation of formyl-peptide PAMPs in the phagolysosome, and their subsequent release into the extracellular space through eructophagy. Furthermore, this process is regulated by cytokines with important, but opposing, roles in the regulation of macrophages in immune responses. Together, these data demonstrate using two unrelated PAMPs that macrophages can activate nearby leucocytes through the release of phagolysosomally processed PAMPs via eructophagy—a process that is upregulated by IFN-γ, and inhibited by IL-4.

## Discussion

Here we describe a cellular process of macrophages which allows for the controlled transient release of soluble matter from mature phagolysosomes directly to the extracellular space—a process which we have termed eructophagy. Initiation and resolution of phagolysosome-plasma membrane communication is mediated in part by gene products associated with degradative and secretory autophagy. Rates of eructophagy vary significantly: from high rates in classically activated macrophages to being virtually absent in IL-4-activated macrophages. Detection of extracellular and intraphagosomal TLR ligands stimulate eructophagy, and newly derived PAMPs released during this process can directly activate leucocytes in the vicinity. We propose that this previously uncharacterized macrophage pathway serves to process and disseminate PAMPs and DAMPs, in response to proinflammatory stimulus, to activate neighbouring and newly recruited leucocytes, and thus serves to amplify local proinflammatory micro-environments within tissues.

The phagosomal membrane, while able to mediate homotypic fusion with late endosomes and lysosomes, is not known to routinely fuse directly with the plasma membrane. Secretory autophagosomes, however, have the ability to fuse with both lysosomes and the plasma membrane[27,28,41,42]. Our findings implicate the autophagy-related proteins Beclin 1, ATG5, ATG7, Atg16L, TSC1 and Pi3K in mediating eructophagy. Notably, Rubicon and Cybb were not required for eructophagy indicating that eructophagy and LAP[25] are mechanistically distinct. Further, while many factors differentiate eructophagy from secretory autophagy, we show that eructophagy utilizes a series of proteins that are involved in fusion/fission machinery associated with secretory autophagy (Supplementary Figure 11)[31]. This leads us to propose a model wherein the phagolysosome could fuse directly with the plasma membrane through the recruitment of key autophagy proteins. Alternatively, eructophagy could be mediated by a two-step process—with the initial recruitment and fusion of autophagosomal membranes with the phagolysosome (creating the ATG5-positive blebs observed on phagolysosomes prior to eructophagy); followed by fusion of the autophagosomal bleb with the plasma membrane serving as channel between phagolysosomal lumen and the extracellular space. In the latter model, phagolysosomal docking and fusion could occur following the recruitment of Rab7 effectors, such as, the HOPS complex, to mediate the assembly of a SNARE fusion complex between Syntaxin 17 on an autophagosomal membrane, Vamp8 on the phagolysosome and Snap29 recruited from the cytosol[28,43,44]. Subsequent fusion between blebbing phagolysosomes with the plasma membrane could then be mediated by the interaction of autophagosomal Sec22b with SNAP23 and Syntaxins 3/4a on the plasma membrane[27,31]. Fission of the hybrid vesicle with the plasma membrane would result in termination of the eructophagy event and dissipation of the autophagosomal components across the phagolysosome. This model is further supported by the rapid

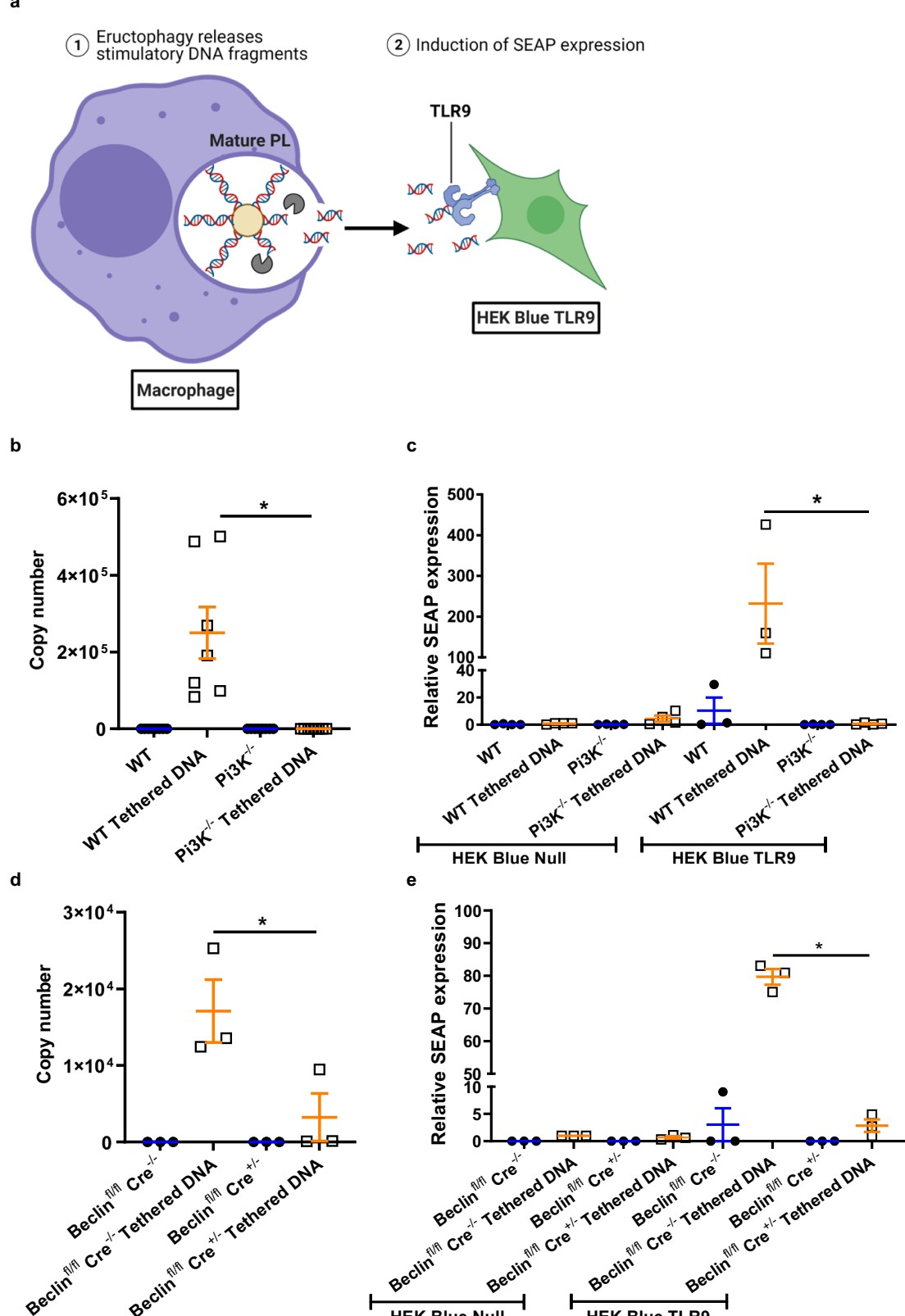

dissemination of LC3-GFP consistently observed across the phagolysosomal membrane during and after the phagolysosomal opening to the extracellular space. Given the known promiscuity of many of these fusion/fission proteins, there are likely redundant or alternative pathways that could be mediating eructophagy and are under further investigation.

A key finding presented in this is the upregulation of eructophagy by pro-inflammatory stimuli, specifically IFN-γ and TLR ligands. IFN-γ has long been known to induce autophagy gene expression in macrophages—a relationship that underlies enhanced resistance of IFN-γ-activated macrophages to intracellular pathogens[45]. Primarily, IFN-γ enhances autophagy

**Fig. 9 CpG DNA released during eructophagy activates vicinal cells. a** Schematic representation of experiment. DNA-conjugated experimental particles are phagocytosed, and particle-restricted DNA must be cleaved by lysosomal DNases prior to eructophagy-mediated extracellular release in order to activate vicinal HEK Blue TLR9 reporter cells through the TLR9-driven expression of SEAP (see text for details). **b, d** Detection of CpG-containing DNA fragments released into the supernatant by **b** PI3K$^{-/-}$ or WT, or **d** Beclin 1 Cre$^{+/-}$ or Beclin 1 Cre$^{-/-}$ BMMØs 4 h after phagocytosis of experimental particles covalently tethered to CpG-containing, 5.1 kb, plasmid DNA or BSA alone. Data are presented as copy number of plasmid DNA as detected by qPCR using a standard curve of CpG-containing, 5.1 kb, plasmid DNA at known concentrations. **c, e** Relative SEAP expression in HEK Blue Null cells or HEK Blue TLR9 cells co-cultured with either, **c** PI3K$^{-/-}$ or WT or **e** Beclin 1 Cre$^{+/-}$ or Beclin 1 Cre$^{-/-}$ BMMØs containing experimental particles covalently tethered to CpG-containing, 5.1 kb, plasmid DNA or BSA alone. Data are presented relative to the negative control HEK Null cells co-cultured with WT BMMØs containing plasmid DNA-tethered-experimental particles. **b** n = 7. **c** n = 3 or n = 4. **d** n = 3. **e** n = 3. Error bars presented as means ± SEM. *P ≤ 0.05 by one-way ANOVA with Bonferroni comparison of means. Source data are provided as a Source Data file.

through the activation of the p38 MAPK signalling pathway, although alternate mechanisms have been proposed involving interferon-regulatory factor 1 (IRF1) and IRF8[46–48]. In the promotion of eructophagy by exposure to TLR ligands, TLR signalling could be operating through the upregulation of autophagy genes, or directly through crosstalk between TLR and autophagy pathways. One proposed direct pathway involves the liberation of Beclin 1 from Bcl2 following activation of MyD88 and TRIF, allowing Beclin 1 to complex with Vps34 and the initiation of eructophagy[49]. Nonetheless, it is clear that proinflammatory stimulus induces eructophagy pointing to a role in inflammation for this cellular process.

To date, innate pattern recognition of microbial products has largely been considered as a consequence of passive release of PAMPs directly from microbes. However, the majority of PAMPs require some level of processing to become freely diffusible and in some cases their partial hydrolysis is a prerequisite for detection by pattern recognition receptors[9–11]. Here we propose that macrophages have evolved to process and utilize PAMPs and DAMPs as secondary signalling molecules that can be strategically deployed to amplify local inflammatory responses. In contrast, in post-inflammatory, repair-focused environments, IL-4-treated macrophages would fully sequester and degrade PAMPs and DAMPs to limit bystander activation. While this study focuses on the release of newly generated PAMPs from exogenous sources, intuitively, eructophagy would also result in the release of endogenous phagolysosomal constituents including proteases, glycosidases, lipases, nucleases, exosomes and antimicrobial peptides. The extracellular release of these endogenously generated lysosomal components may also have effects on the tissue microenvironment during inflammation. For instance, the release of lysosomal proteases may act to degrade the extracellular matrix and activate protease-activated receptors in neighbouring cells[50,51]. Eructophagy may additionally disseminate endogenous or exogenous antigens which could in turn allow for expanded MHC-restricted presentation of oligopeptides by other antigen-presenting cells, enhanced co-stimulation of T cells, augmented humoral immune responses in draining lymph nodes, and potentially trigger local mast cell degranulation through IgE. Eructophagy could also be potentially co-opted by pathogens to release immunomodulatory virulence factors, toxins, or enhance viral dissemination. Indeed the observed release of immunomodulatory Mycobacterial lipids from *Mycobaterium* spp.-infected macrophages may be facilitated by eructophagy, potentially contributing to granuloma formation in tuberculosis[52–54]. Together, in addition to PAMP processing and release, eructophagy may have numerous biologically relevant consequences that have yet to be identified and explored.

The future study of eructophagy presents some technical challenges. The ephemeral and stochastic nature of eructophagic events—with only a fraction of a percent of phagolysosomes

undergoing eructophagy at any point in time—requires the use of highly time-resolved, high-content live microscopy and generally precludes static approaches such as immunofluorescence microscopy, flow cytometry or electron microscopy as traditionally used to study phagosomes and autophagosomes. Additionally, the difficulty of genetically manipulating primary macrophages coupled with the stark absence of eructophagy in traditional phagocytic cell lines such as J774, RAW 264.7 and DC2.4 necessitated the use of primary cells or conditionally immortalized GB8 macrophages. The predominance of the conveniently mutable macrophage-like cell lines in the study of phagocyte cell biology, coupled with the highly transient nature of eructophagy, most likely contributed to obscuration of this phenomenon to the field hitherto. Nonetheless, despite the lack of current understanding of this previously undescribed phagolysosomal process, the utilization of precursor-derived macrophages and live microscopy together with the methods developed as a part of this study will enable the future investigation of eructophagy to occur at an ever-increasing pace.

Traditionally the macrophage is considered to have two critical, but functionally separate roles: first, the coordination of inflammation and repair through paracrine communication to vicinal cells; and second, the removal of microbes and dead cells through phagocytosis and degradation within the phagolysosome. Here propose that eructophagy allows macrophages to process and disseminate phagocytosed PAMPs and DAMPs to amplify local inflammatory responses. By utilizing phagocytosed material to manipulate the pro-inflammatory state of the local microenvironment, eructophagy is an eloquently simple collaboration between these two traditional roles of the macrophage.

## Methods

All animal use, care and husbandry were conducted according to protocols approved by respective Animal Care and Use Committees at the University of Calgary or Washington University, St. Louis. Human samples were utilized as approved by the Conjoint Health Research Ethics Board at the University of Calgary.

**Mice.** C57BL/6J (WT), TSC1$^{fl/fl}$ [55], B6.129S6-Cybb$^{-/-}$ (Cybb$^{-/-}$)[56], MyD88$^{-/-}$ [57], PI3Kγ$^{-/-}$ [58] and B6.129P2-Lyz2 mice (lysM-Cre) were purchased from The Jackson Laboratory (Bar Harbor, ME, USA) and maintained at the University of Calgary, AB, Canada under standard conditions. Beclin 1$^{fl/fl}$ were provided by Dr. Edmund Rucker (University of Kentucky, KY, USA) and MyD88/TRIF$^{-/-}$ mice were provided by Dr. Paul Kubes (University of Calgary, Canada) and maintained at the University of Calgary, AB, Canada under standard conditions. ATG5$^{fl/fl}$ [59], ATG16L$^{fl/fl}$ [60], ATG7$^{fl/fl}$ [61], Rubicon$^{-/-}$ [25] and LC3-GFP[62] mice were maintained at Washington University, St Louis. Where indicated, LoxP-flanked target strains were crossed with LysM-Cre strains[60] for conditional knockout of LoxP-flanked targets in bone marrow-derived macrophages (BMMØs). Homozygous LoxP-flanked, LysM-Cre-negative littermates were used as controls in all experiments utilizing LysM-Cre-positive mice. All mice were used between 8 and 12 weeks, we did not see any sex differences in our studies so males and females were used interchangeably in experiments.

**Cell preparation.** BMMØs and bone marrow-derived DCs were derived from bone marrow extracted from 8- to 12-week-old mice using L929- and GMCSF-conditioned media, respectively, as previously described[15,63]. Murine peritoneal

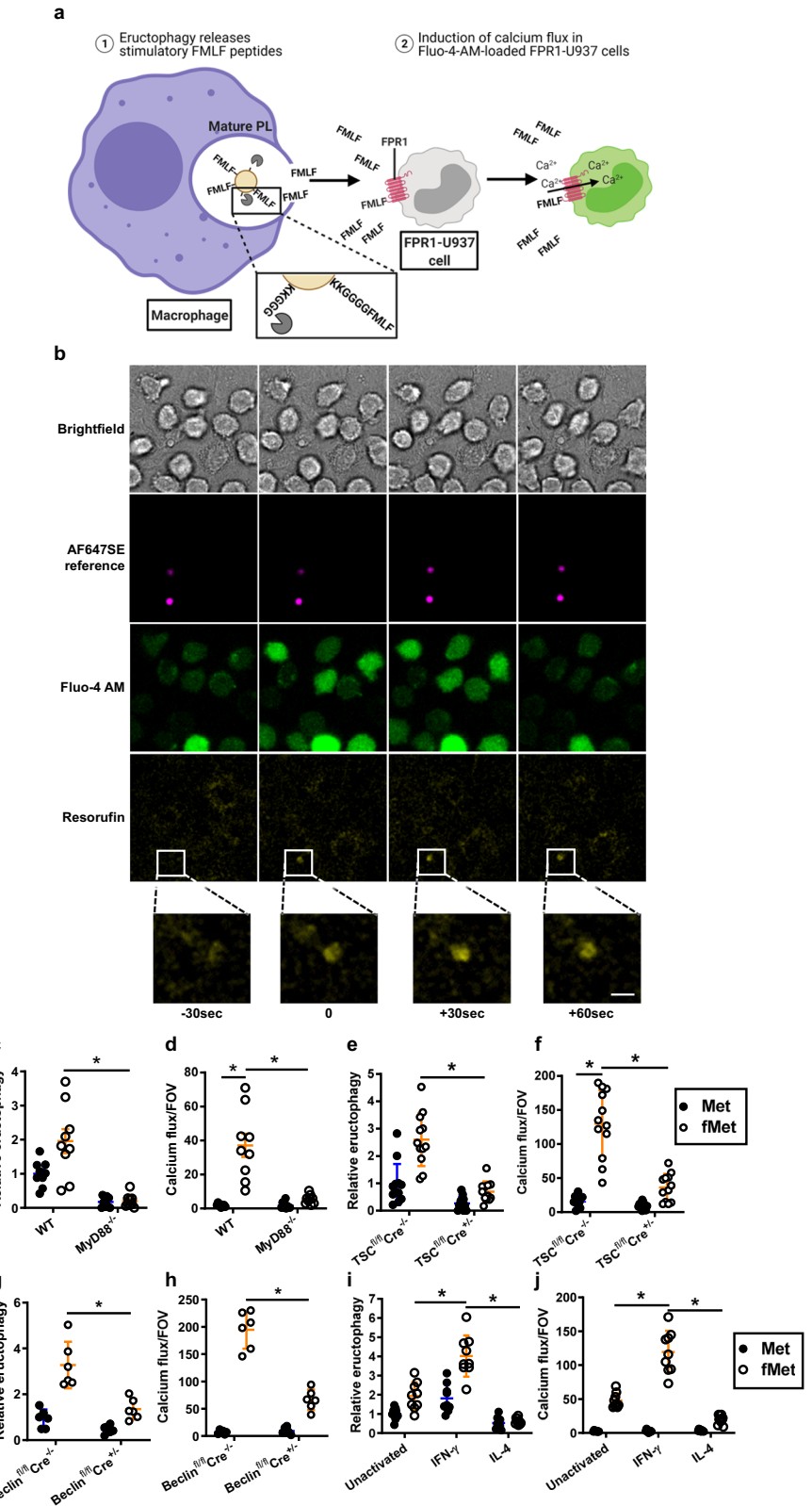

macrophages were isolated from euthanized 8- to 12-week-old mice by peritoneal lavage with cold Dulbecco's phosphate-buffered saline (DPBS) pH 7.2 without elicitation. Splenic CD11b+ macrophages were isolated from 8- to 12-week-old mice using a CD11b-positive selection kit (Stem Cell Technologies) as per the manufacturer's instructions. Human CD14+ peripheral blood monocytes (PBMs) were isolated from the peripheral blood mononuclear cell population of healthy human volunteers (as approved by the Conjoint Health Research Ethics Board at the University of Calgary), by positive selection using anti-CD14 paramagnetic

microbeads, according to the manufacturer's instructions (Miltenyi Biotec). PBMs were subsequently plated and cultured for 5 days in the presence of 10% foetal bovine serum to drive differentiation to an adherent macrophage phenotype. Conditionally immortalized myeloid precursors were generated through transduction of myeloid precursors with the pMSCV-ER-HoxB8 retroviral plasmid[18]. Cells were maintained in their myeloid precursor state through use of 2 μM β-estradiol. To induce differentiation, cells were incubated in L929-conditioned media without β-estradiol, in the presence of the oestrogen receptor antagonist

**Fig. 10 Formylated peptides released during eructophagy activate vicinal cells. a** Schematic representation of experiment. Following phagocytosis of reporter particles with tethered formylated (fMet) or non-formylated (Met) peptides, BMMØs were co-cultured with formyl peptide receptor 1 (FPR1)-expressing U937 cells loaded with the calcium-sensor Fluo-4 AM. Particle-restricted peptides must be cleaved in the phagolysosome in order to be released during eructophagy. **b** Sequential confocal images demonstrating temporal and spatial relationships between an eructophagic event (yellow) in WT BMMØs, as detected by AF647-labelled reporter beads (magenta) conjugated to cellulase and fMet peptides, and the associated cytosolic calcium flux (green) in vicinal FPR1-U937 cells, triggered by extracellular release of fMet peptides by the BMMOs. **c–h** Rates of eructophagy in BMMØs containing reporter particles with tethered formylated (fMet) or non-formylated (Met) peptides, and the corresponding activation of FPR1 in neighbouring U937 cells in the same FOV. **c**, **e**, **g**, **i** Relative rates of eructophagy (as detected by phagolysosomal pH) in **c** MyD88$^{−/−}$ BMMØs relative to WT BMMØs, **e** BMMØs conditionally deficient in TSC1 (Cre$^{+/−}$) relative to BMMØs derived from Cre negative littermates (Cre$^{−/−}$), **g** BMMØs conditionally deficient in Beclin 1 (Cre$^{+/−}$) relative to BMMØs derived from Cre negative littermates (Cre$^{−/−}$), and **i** WT BMMØs with prior exposure to IFN-γ (100 U/mL, 18 h) or IL-4 (10 ng/mL, 24 h) relative to unstimulated WT BMMØs. **d**, **f**, **h**, **j** The corresponding incidence of calcium fluxes in co-cultured FPR1-U937 cells per FOV. Calcium fluxes are presented as a numerical average of fluxes detected over ≥2 FOV. **c–d** $n = 9$ and **g**, **h** $n = 6$ FOVs over three biologically independent experiments (**e**, **f** and **i**, **j**) $n = 11$ FOVs over four biologically independent experiments. Error bars presented as means ± SEM. *$P ≤ 0.05$ by two-way ANOVA with Bonferroni comparison of means. Time 0 represents an eructophagy event. Scale bars denote 5 μm. Source data are provided as a Source Data file.

fulvestrant (1 μM). After ten days cells terminally differentiated into GB8 macrophages. pMSCV-ER-HoxB8 retroviral plasmid was a kind gift from Dr. David Sykes (Harvard University, MA, USA).

For imaging, cells were seeded in 96 well μClear® plates (Greiner Bio-One) or Nunc$^{TM}$ MicroWell$^{TM}$ optical bottom 96-well plates (Thermo Fisher Scientific) and allowed to establish a confluent monolayer 16–24 h prior to evaluation by high content or scanning confocal microscopy, respectively. For immunofluorescence analysis, cells were seeded on a 12-mm microscope cover glass (VWR) in a 24-well plate overnight. Where indicated, BMMØs were cultured in the presence of either 10 ng/mL recombinant murine IL-4 or 100 U/mL recombinant murine IFN-γ (PeproTech) for 24 and 18 h respectively. Where indicated, cells were pulsed overnight with 2 mg/mL Bella Fluor (BF) 350SE-labelled (Setareh Biotech) dextran (70 kDa) with the dextran being chased into lysosomes the following day for 2 h. Where indicated, 2.5 mM of LLOMe was added to cells 30 min after phagocytosis of experimental particles and incubated for 2 h prior to imaging. Where indicated, 10 nM of myricetin (Alfa Aesar), 10 nM of kaempherol (Selleckchem) or 100 nM of EACC (Life Chemicals) were added to cells at the same time as experimental particles. Where indicated 0.5 μM Torin 1 (Tocris Bioscience), 0.5 μM rapamycin (Sigma Aldrich), 1× MEM amino acid solution (Sigma Aldrich), 10 ng/mL ultrapure lipopolysaccharide from *S. Minnesota* R595 (LPS; List Biologicals) were added to assay medium 30 min after the addition of reporter particles or 50 nM Lysotracker® Red (Invitrogen) 2 h after the addition of experimental particles and remained in the assay medium for the duration of phagosomal assessment.

**Single phagosome assessment using high content imaging**. To monitor eructophagy, pH-sensitive fluorophores, fluorogenic substrates or enzymes, along with Alexa Fluor® 647-succinimidyl ester (AF647SE) as a reference fluor, were covalently coupled to experimental particles. To simultaneously monitor phagosomal proteolysis and pH, 3.0 μm carboxylate-functionalized silica particles (Kisker Biotech) were coupled to the fluorogenic protease substrate DQ Green BSA (Invitrogen) and human IgG (Sigma Aldrich) using the heterobifunctional crosslinker, cyanamide (Sigma Aldrich), as previously described[16]. After quenching any unreacted cyanamide, the particles were subsequently labelled with the pH-sensitive reporter pHrodo™ succinimidyl ester (pHrodoSE) (Invitrogen) and the reference fluor, AF647SE. Enzyme-coupled reporter particles were synthesized by the covalent conjugation of human IgG (Sigma Aldrich) and α-amylase from *Bacillus licheniformis* (LD Carlson Co.), cellulase from *Trichoderma viride* (Cellulysin®; EMD Millipore), or β-glucosidase from almonds (Sigma Aldrich) to 3.0 or 5.0 μm carboxylate-modified silica (Kisker Biotech) or 4.5 μm carboxylate-modified polystyrene (Bangs Laboratories) particles using the heterobifunctional crosslinker cyanamide (Sigma Aldrich) as described above. To monitor changes in pH of phagolysosomes containing bacteria and yeast, mid-log *Staphylococcus aureus*, *Vibrio parahaemolyticus* or *Saccharomyces cerevisiae* were fixed with 4% paraformaldehyde and labelled with CFSE, pHrodoSE and AF647SE through an ethylenediamine linker introduced via reductive amination. Since the enzymes used for reporter particles contained trace amounts of Toll-like receptor (TLR) ligands, for experiments investigating phagosomal and plasma membrane TLR activation and release of fMLF, experimental particles coated with PAMP-free defatted BSA labelled with CFSE or pHrodoSE and AF647SE were synthesized using cyanamide as described above. Where indicated, the TLR agonists polyinosinic:polycytidylic acid (5 μg/mL, Poly I:C; Invivogen) and unmethylated plasmid DNA (100 μg/mL, isolated from *Escherichia coli* using EndoFree Plasmid Maxi Kit, Omega Bio-tek) were non-covalently adsorbed onto the surface of the defatted BSA-coupled particles with gentle sonication. Excess PolyI:C or DNA was removed through extensive washing of the experimental particles prior to their addition to BMMØs.

Experimental particles were added to cells in an assay medium (DPBS pH 7.2 supplemented with 1 mM CaCl$_2$, 2.7 mM KCl, 0.5 mM MgCl$_2$, 5 mM glucose,

10 mM HEPES and 2.5 g/L gelatin from porcine skin Type A) to achieve a target of 2–3 beads/cell, 2–3 yeast/cell or 8–10 bacteria/cell. Detection of eructophagy using enzyme-coupled reporter particles was achieved by the addition of the membrane-impermeable fluorogenic substrates, resorufin maltotriose, resorufin cellobioside, or resorufin glucopyranoside (Marker Gene Technologies) at 8.2 μg/mL to assay medium 60 min following phagocytosis of the α-amylase, cellulase or β-glucosidase reporter particles, respectively.

High content imaging was undertaken using the IN Cell Analyzer 2000 v. 5.2 (GE Lifesciences) automated microscope equipped with an environmentally controlled chamber (set to 37 °C, 5% CO$_2$) as previously described[16]. Images were acquired every 1–3 min over a period of 2–10 h. Single phagosome tracking and fluorescent quantification were carried out using the IN Cell Investigator 1.8.3 software was used to analyse data in conjunction with Spotfire DecisionSite® Client software (TIBCO Software Incorporated) and a custom designed analysis app based on R[16]. Eructophagy was quantified as either a percentage of the number of phagolysosomes that underwent at least one eructophagy event as a factor of the total number of phagolysosomes in a field of view, or where indicated, the total number of eructophagy events in a field of view as a factor of the total number of phagolysosomes over the indicated period of time. Data are representative of the average of at least three randomly selected fields of view across at least two wells. Confocal microscopic images and videos were acquired using a Leica SP5 confocal microscope equipped with a HyD hybrid photon detector, HCS scanning stage and environmental control set to 37 °C (Leica Microsystems).

**Targeted knockdown in conditionally immortalized cells**. Conditionally immortalized macrophage precursors were transduced with select lentiviral particles chosen from the MISSION shRNA TRC2 lentivirus library (Sigma Aldrich). These cells were pretreated for 1 h with 8 μg/mL polybrene prior to spinoculation at 1500 × *g* for 1 h at 32 °C. Cells were allowed to recover overnight before selection with 20 μg/mL puromycin for 4 days. Verification and analysis of the level of gene knockdown were carried out using quantitative PCR (qPCR) for the gene of interest as described below and previously[64].

**CpG DNA conjugation and HEK Blue cell co-culture**. CpG-containing, 5.1 kb, plasmid DNA (pDsRedC1-ODNx15HindIII-BamHI), was crosslinked to 3.0 μm silica particles bearing defatted BSA using 1 mg succinimidyl-[4-(psoralen-8-yloxy)]-butyrate (SPB) in PBS with exposure to UV light >350 nm for 30 min. Particles were thoroughly washed with pre-warmed assay medium before addition to BMMØs. Forty-five minutes after particles addition, monolayers were washed thoroughly to remove non-phagocytosed particles prior to the addition of HEK Blue Null or HEK Blue TLR9 (Invivogen, hkb-null1 and hkb-htlr9, respectively) reporter cells at 80% confluency. Supernatants and cell pellets were collected after 4 h for qPCR analysis as described in the section below.

**Quantitative polymerase chain reaction**. qPCR was used to measure relevant transcript levels from total RNA isolated from Beclin 1$^{fl/fl}$, ATG16L$^{fl/fl}$, ATG5$^{fl/fl}$, ATG7$^{fl/fl}$ and TSC1$^{fl/fl}$ lysM-Cre positive and Cre negative BMMØs, Rab5a$^{kd}$, Vps41$^{kd}$, Vps39$^{kd}$, Sec22b$^{kd}$, Vps33a$^{kd}$, Stx3$^{kd}$, Elastase2$^{kd}$, Stx17$^{kd}$, Rab7$^{kd}$, Snap23$^{kd}$, Stx4a$^{kd}$, Snap29$^{kd}$, Stx1b1$^{kd}$, Stx1b2$^{kd}$, Exoc8$^{kd}$, Snapap$^{kd}$, StxBP5l$^{kd}$, Rab28$^{kd}$, Rab34$^{kd}$, Vamp8$^{kd}$, Snap23$^{kd}$, Stx18$^{kd}$ and WT control GB8 macrophages on the day of experimental use following standard protocols[64]. RNA was isolated using Trizol (Invitrogen) as per the manufacturer's protocol[64]. Primers were designed using the Roche Universal ProbeLibrary Assay Design Center and are as follows; *Tsc1* F: CGGGTCGTTCAGACAACTGA, R: CAACTGACCTGGGTGA AACC; *Becn1* F: AGGATGGTGTCTCTCGAAGATT, R: GATCAGAGTGAA GCTATTAGCACTTTC; *Atg7* F: CAAAGGCTTTCACCAAAACAG, R: TCGGCT CGACACAGATCAT; *Atg5* F:AAGTCTGTCCTTCCGCAGTC, R:TGAAGAAAG

TTATCTGGGTAGCTCA; *Rab5a* F: AGCTCCAGTGTGAACTTCCT, R: ACTG-GAGATGAGCTGTTCCC; *Vps41* F: CCAAGGAACGAGACCAGGAT, R: CTTGCAGCCATGTCGTACTC; *Vps39* F: TCTTCCCTTGTACCGTCCAC, R: ATCCTGCCTAGACCACACAC; *Sec22b* F: TGAAGAAGTCCTACAGCGGG, R: CCACCAAAACCGCACATACA; *Vps33a* F: AAGGGATGATGAACTGGGCA, R: TGAATTCGTCCCCTCAGCT; *Stx3* F: GGGAACAAGTCTCTGAGCCC, R: GTGGTCTGTCTGGCTTCGAT; *Ela2* F: TCTCTGTGCAGCGGATCTTC, R: TGATGGTCTGTTTGTGCCCA; *Stx17* F: GCGGGAGGTGTTTCTGTTTG, R: GCACCGCTGATACTTCTCGA; *Rab7* F: AGTTTCTCATCCAGGCCAGC, R: CTTGGCACTGGTCTCGAAGT; *Snap23* F: AGAACTGTGGAGGCTGGAGA, R: TGTTCCCCTTGCTCATCCAG; *Stx4a* F: TGTGGATGGTGAAAGGGGTG, R: CTCAACTCGTGGGTCCTGTC; *Stx1b* F: AAGGATCGGACTCAGGAGCT, R: GTCTTCTCATCGGGGTTGGG; *Stx18* F: CGCGTGGTGGACAAGAAAAG, R: AGCTCATCTTCACCCTTGCC; *Elastase2* F: TCTCTGTGCAGCGGATCTTC, R: TGATGGTCTGTTTGTGCCCA; *Snap29* F: AGCCACCCAAACCTCAGAAG, R: TCCCACGGACAGCTCATCTA; *Atg16L* F: GCAGCTACTAAGCGACTCTCG, R: AGACAGAGCGCTCTCCCAAAG; and *18s* F: CGCGGTTCTATTTTGTTGGT, R: AGTCGGCATCGTTTATGGTC. For detection of CpG-containing, pDsRedC1-ODNx15HindIII-BamHI plasmid DNA from supernatants, the following primers were used: F: GTGACCGTGACCCAGGACTC and R: CGCCGATGAACTT-CACCTTG. SEAP gene expression in Hek Blue cells was detected using F: CCCACCTTGGCTGTAGTCAT and R: TCTGGGTACTCAGGGTCTGG. qPCR was carried out in a Bio-Rad iQ5 thermocycler using iQ SYBR Green Supermix (Bio-Rad) according to the manufacturer's instructions[64]. Target gene expression was calculated relative to 18S rRNA and to corresponding WT, Cre-negative samples or shRNA knockdown controls. Plasmid copy numbers were calculated by qPCR using a standard curve of parental plasmid DNA as a template.

### *N*-formyl peptide conjugation and BMMØ-U937<sup>FPR-1</sup>+ co-culture.

*N*-formylated and non-*N*-formylated peptides (fMLFGGGGKK and MLFGGGGKK, respectively) were custom-synthesized by Synpeptide (Shanghai, China) and covalently coupled to 3.0 μm silica particles bearing defatted cBSA or Cellulysin® using 1-ethyl-3-(3-dimethylaminopropyl) carbodiimide (EDC) and *N*-hydroxysulfosuccinimide (sulfo-NHS). These particles were further labelled with pHrodoSE and AF647SE or AF647SE alone, quenched with glycine, and thoroughly washed before addition to BMMØs. Where indicated WT BMMØs were cultured in the presence of either 10 ng/mL IL-4 or 100 U/mL IFN-γ for 24 and 18 h prior to assay respectively. To image cytosolic calcium fluxes elicited by detection of N-formyl peptides in neighbouring cells, U937 cells stably expressing human FPR-1 [65] (a gift from Dr. Eric Prossnitz, University of New Mexico) were loaded with the calcium indicator Fluo-4 AM (5 μg/mL) for 30 min (Setareh Biotech) in calcium-free HBSS and then added to washed BMMØs 45 min after phagocytosis of the peptide-decorated particles. To investigate the temporal and spatial relationship between eructophagy and *N*-formyl-peptide detection by neighbouring cells, the cellulase substrate resorufin cellobioside was added to the medium prior to sequential confocal imaging with 30 s intervals. To determine the association between the frequency of eructophagy of macrophages and FPR-1-elicited calcium fluxes in co-cultured FPR-1-U937 cells, eructophagy was induced through the addition of 10 ng/mL LPS to all BMMØs prior to imaging and was performed using the IN Cell Analyzer 2000 HCS microscope.

### Lysosomal markers.

To monitor LAMP1 on the phagolysosomal membrane during eructophagy, mEmerald-tagged rat LAMP1 was expressed in BMMØs (mEmerald-Lysosomes-20 plasmid was a gift from Michael Davidson, Addgene #54149). BMMØs were transfected using the Nucleofector™ kit for mouse macrophages (Lonza Group) and the Y-001 electroporation programme as per the manufacturer's guidelines. Transfected BMMØs were seeded overnight in complete medium prior to assessment. To observe interactions of lysosomes and phagolysosomes following eructophagy, the fluorescent acidotropic probe LysoTracker® Red (50 nM, Invitrogen) was added to BMMØs 2 h following addition of experimental particles bearing the protease substrate DQ Green BSA. Where indicated, lysosomes of BMMØs were loaded with Alexa Fluor® 594 hydrazide or 150 kDa dextran labelled with Alexa Fluor® 488 by overnight incubation with the soluble fluorescent molecules followed by a 2 h chase period prior to imaging as previously described[15].

### Immunofluorescence.

Experimental particles conjugated to either DQ Red BSA or DQ Green BSA, the reference fluor AF647SE and microbial PAMPs (where indicated) were added to GB8-derived macrophages or BMMØs for 3 h prior to fixation with 4% formalin/PBS. Cells were permeabilized in ice-cold PBS containing 1 mg/mL Zwittergent 3-12 (Millipore) and blocked in PBS containing 2.5% serum corresponding to the host species of the secondary antibody used. Cells were probed with rabbit anti-ATG5 (Proteintech, 10181-2-AP) diluted 1:100 in blocking buffer at room temperature overnight. After cells were incubated with a goat anti-rabbit secondary antibody conjugated to AF488 (Thermo Fischer Scientific) diluted 1:500 in blocking buffer for 1 h. Confocal microscopy images and videos were taken using a Leica SP5 confocal microscope equipped with a HyD hybrid photon detector, HCS scanning stage (Leica Microsystems). Super resolution images were acquired using a Leica SP8 Lightning system. Image analysis and quantification were carried out using Image J software with manual enumeration.

### Proteolytic activity and bead uptake analysis.

Real-time measurement of intraphagosomal proteolysis as a proxy for phagosomal maturation and function was performed as previously described[66]. In brief, 3.0 μm experimental particles bearing human IgG (Sigma Aldrich) and the fluorogenic protease substrate DQ Green BSA (Invitrogen) labelled with the c fluor, AF594SE (Invitrogen) (as described above) were added to cells to achieve a target of 2–3 beads/cell. Phagosomal hydrolytic activity was evaluated by measuring the rate of substrate-liberated fluorescence relative to the calibration fluor. Relative fluorescent units (RFU) as described by the equation $RFU = SFRT/CFRT$ (where SFRT is the substrate fluorescence in real time and CFRT the calibration fluorescence in real time) were plotted against time. Measurements were performed in microplate format using a FLUOstar Optima Fluorescent plate reader (BMG Labtech) at 37 °C. For the evaluation of phagocytic uptake, cells were incubated with 3.0 μm experimental particles bearing human IgG (Sigma Aldrich) and BSA labelled with Oregon Green 488 SE (Thermo Fisher) for 60 min prior image analysis using the IN Cell Analyzer 2000 (GE Lifesciences) automated microscope. Phagocytosis was evaluated by quantifying the number of experimental particles engulfed per cell with the aid of Image J software.

### Quantification and statistical analysis.

GraphPad Prism® versions 5.0, 8.0 and 9.0 (GraphPad Software) was used for all statistical analysis. For analysis of a single variable between two samples, Student's unpaired *t*-test was used. For analysis of a single variable between three or more samples, one-way ANOVA with Bonferroni's multiple comparison post hoc test was used. For comparisons of two or more variables in two or more samples, two-way ANOVA with Bonferroni's multiple comparison post hoc test was used. Data were considered statistically significant when $P \leq 0.05$.

**Reporting summary.** Further information on research design is available in the Nature Research Reporting Summary linked to this article.

## Data availability

The datasets generated during and/or analysed during the current study are available from the corresponding author (rmyates@ucalgary.ca) on reasonable request. All source data are available in the Source Data file.

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

## Acknowledgements

We thank Dr. J. Canton, University of Calgary, for his sage advice. All graphical illustrations and figures were created with BioRender.com.

## Author contributions

R.M.Y., C.J.G., J.A.N., S.M.C., C.R.A. and D.R.B. designed the experiments; C.J.G., J.A.N., S.M.C., C.R.A., D.R.B., N.M., Y.T.W., D.A., A.S. R.I.C. and B.W.E. performed the experiments and analysed the data. H.W.V. provided reagents and intellectual contribution.

## Competing interests

The authors declare no competing interests.
