## [Peer Review File · Nature Communications]

Macrophages disseminate PAMPs through the direct extracellular release of the soluble content of their phagolysosomesREVIEWER COMMENTS

Reviewer #1 (Remarks to the Author):

Greene et al in the paper entitled "Direct extracellular release of phagosomally-processed PAMPs by macrophages" report a study demonstrating a new cellular process called Eructophagy. Using a series of imaging techniques, they tried to demonstrate that phagolysosomes can transiently fuse with cell membrane and release their partially digested content extracellularly. They also showed that autophagy machineries are required for Eructophagy. Lastly, they demonstrated that Eructophagy is required for releasing active DMAPs/PAMPs. The major strength is the novelty of this discovery and potential biological significance of this cellular process. The experimental results are convincing and largely supportive of their main conclusion. Although many elegant assays have been employed, it would be ideal to show Eructophagy by electron microscopy. In addition, at the molecular level, the authors should clarify how Eructophagy differs from autophagy-mediated secretion.

Specific critics:

Fig2. Panel e. the legend is quite confusing. In the text, the authors stated that some phagolysosomes undergo several eructophagy events. This figure does not seem to show that. In addition, the method section describing how these events are quantified is quite vague. In b and e only one trace is shown. It would be informative to show accumulative data from many phagolysosomes.

Fig3e. The figure legend is confusing. There is not time zero in this panel.

For Fig5a, Whether the deletion of autophagy genes affects the macrophage differentiation should also be clarified. In addition, it should be clarified whether autophagy deficiency affects the number and dynamics of phagosomes. Fig5c. the use of this dye has generated confusing results. First the MOA of this dye is not clear; second, in terms of kinetics, the results are in conflict with Fig5d, which is a more reliable assay for autophagy-related molecular events. It is not explained why LC3-GFP started to accumulate after eructophagy not before.

Fig9 and 10. It would be more convincing to show that the autophagy-deficient macrophages (such as beclin 1 or Atg7 deficient macrophages) fail to release PAMPs.

Minor points:

"During eructophagy, the phagolysosomal membrane retained LAMP1, and phagolysosomes were able to fuse with lysosomes immediately following an eructophagic event (Supplemental Fig. 6a, b" should have been suppl Fig5.

Reviewer #2 (Remarks to the Author):

This manuscript attempts to coin a new term around a process in which autophagic machinery allows for the extracellular release of phagocytic cargo to stimulate surrounding cells with PAMPs, DAMPs, or in theory antigens. This underlying concept is very innovative, even provocative. However, the data in the manuscript fell short of demonstrating the concept and the quality of the data made analyzing the conclusion challenging. There were some innovative approaches incorporated but greater rigor at many levels is needed, including in the data presentation. Overall, there are several major shortcomings.

1. on page 10, the authors conclude that the loss of fluorescence associated with the vesicles that undergo transient loss of acidification is not due to peptides moving into the cytoplasm so it is likely instead that the peptides are deposited extracellularly. However, data for neither point (fluorescence

of the cytoplasm or analysis of extracellular peptides) is shown/cited. This problematic overinterpretation continues throughout the manuscript.

2. In Fig 3, a key assay with resorufin is developed which will be used for much of the remaining manuscript. The results are used to conclude that because resorufin had access from the outside to the vesicles it must mean that they were fusing and transiently available to the extracellular space. However, isn't there an alternative possibility? The alternative seems to be that resorufin gets taken into vesicles intracellularly and that these then fuse with phagolysosomes that led to substrate conversion? How do the authors rule that out?

3. Later, using the resorufin assay, the authors demonstrate that "eructophagy" is dependent upon Beclin 1. This result is critical to the novelty of the whole story. However, later, when they set up experiments in later figures (Fig 9, 10), they do not use the Beclin 1KD as a control. This greatly weakens those figures, because it is impossible in the later figures to be sure that eructophagy is the key mechanism behind the transfer of signals from one macrophage to the other.

4. Throughout the manuscript, there is a low standard for data quality, succinct communication, and transparency with adherence to conventional standards. Specific examples include:

- none of the movies is annotated with arrows and generally fail to communicate the results the authors intended, as the reviewer lacks capacity to orient without processing and annotation.
- there are several parts of the manuscript that have redundant sections. The introduction is conceptually engaging but could and should be greatly reduced in length.
- Surprisingly, unless I missed it, I could not find a listing of author contributions/roles. No grant funding or other financial sourcing was listed. Some authors are listed with past affiliations. It may be appropriate to list Herbert Virgin, for instance, as being affiliated with his former university of employment if that is where he was when he contributed to the work in whatever role he played. However, it would also be important in the interest of transparency and convention to list his present institution in the acknowledgements. Indeed, the Acknowledgements section was, in contrast to other sections of the manuscript, inappropriately brief.

Response to reviewers

We thank both reviewers for their insightful comments and suggestions. We have addressed all comments where possible and it has resulted in a significantly improved manuscript. Thank you.

1. Although many elegant assays have been employed, it would be ideal to show Eructophagy by electron microscopy.

We agree that this study would benefit from electron microscopic imagery/data. However, since this is a transient (seconds/minutes), hyper-localized, relatively infrequent event, the chances of capturing the phagolysosomal communication with the plasma membrane with a static approach such as TEM would be near impossible. Moreover, it would be more likely to misinterpret fixation artifacts as eructophagic events than it would be capture the bona fide process that we have described using dynamic real-time approaches. Indeed, we believe that the overreliance on solely utilizing static approaches to study phagosomal biology—such as immunofluorescent microscopy, TEM, and flow cytometry—has resulted in the obfuscation of eructophagy thus far. The limitations of the study and the difficulty of using traditional approaches to study eructophagy are discussed in the manuscript.

2. In addition, at the molecular level, the authors should clarify how Eructophagy differs from autophagy-mediated secretion.

Thank you for this suggestion. We have added Supplemental Figure. 11 to summarize the known differences between eructophagy and autophagy-mediated secretion

3. Fig2. Panel e. the legend is quite confusing. In the text, the authors stated that some phagolysosomes undergo several eructophagy events. This figure does not seem to show that. In addition, the method section describing how these events are quantified is quite vague. In b and e only one trace is shown. It would be informative to show accumulative data from many phagolysosomes.

This has been addressed in Supplemental Figure. 5 which shows trace graphs of several phagolysosomes undergoing multiple events over the same time period. The figure legends have been altered for clarification. Additionally, we have added more detail and explanations to our methods section to describe the quantification method.

4. Fig3e. The figure legend is confusing. There is not time zero in this panel.

The figure legend of Fig 3 has been altered to better explain the graphs and time.

5. For Fig5a, Whether the deletion of autophagy genes affects the macrophage differentiation should also be clarified. In addition, it should be clarified whether autophagy deficiency affects the number and dynamics of phagosomes. Acidification with no calibration & proteolysis (dynamics) Fig5c. the use of this dye has generated confusing results. First the MOA of this dye is not clear; second, in terms of kinetics, the results are in conflict with Fig5d, which is a more reliable assay for autophagy-related molecular events. It is not explained why LC3-GFP started to accumulate after eructophagy not before.

This has been addressed in Supplemental Figure. 6 which shows the phagosomal kinetics and bead uptake ability of macrophages deficient in some key autophagy genes investigated throughout the paper. Regarding the temporal recruitment of LC3-GFP relative to eructophagy, we too do not know why LC3 accumulates after eructophagy. We speculate in the discussion section that it is involved in membrane fission, but this requires further investigation.

6. Fig9 and 10. It would be more convincing to show that the autophagy-deficient macrophages (such as beclin 1 or Atg7 deficient macrophages) fail to release PAMPs.

This has been addressed in Figure. 9 (d and e). Additional data has been incorporated to show DNA release in Beclin 1 deficient macrophages. Furthermore, data showing the effect of Beclin1 deficiency in FMLF release during eructophagy has been incorporated into Figure 10 (g and h).

7. Minor points:

“During eructophagy, the phagolysosomal membrane retained LAMP1, and phagolysosomes were able to fuse with lysosomes immediately following an eructophagic event (Supplemental Fig. 6a, b” should have been suppl Fig5.)

We apologize for this error and it has been rectified in the revised manuscript.

8. on page 10, the authors conclude that the loss of fluorescence associated with the vesicles that undergo transient loss of acidification is not due to peptides moving into the cytoplasm so it is likely instead that the peptides are deposited extracellularly. However, data for neither point (fluorescence of the cytoplasm or analysis of extracellular peptides) is shown/cited. This problematic overinterpretation continues throughout the manuscript.

This has been addressed in several ways.

- In addition in to the wide-field images in 2c, green cytosolic fluorescent trace has been added to 2d. Both show a considerable loss of green fluorescence (both magnitude and volume) from the phagolysosome, and no corresponding gain in cytosolic fluorescence.
 - Figure. 3 (a-d). This new data demonstrates the loss of phagolysosomal fluorescent dextran at the same time as resorufin signal, without entry of cytosolic GFP during an eructophagic event (Fig 3 a,b). In contrast, rupture of the phagolysosome (induced by LLOMe treatment) demonstrates the leakage of dextran from the phagolysosome at the same time as cytosolic GFP into the phagolysosome. Only when the LLOMe-treated macrophages have necrotized, and the bead-conjugated cellulase is exposed extracellularly, do we see cleavage of the resorufin substrate (Fig 3 c,d).
 - Figure 9 directly (b,d) and indirectly (c,e) measures the loss of phagolysosomally-processed DNA from macrophages to the extracellular medium in eructophagy-proficient and eructophagy-defective macrophages.
9. In Fig 3, a key assay with resorufin is developed which will be used for much of the remaining manuscript. The results are used to conclude that because resorufin had access from the outside to the vesicles it must mean that they were fusing and transiently available to the extracellular space. However, isn't there an alternative possibility? The alternative seems to be that resorufin gets taken into vesicles intracellularly and that these then fuse with phagolysosomes that led to substrate conversion? How do the authors rule that out?

Although possible that pino/endocytic vesicles can fuse with the phagolysosome, we have ruled this out as we observe resorufin 'flashes' immediately after addition of this substrate into the extracellular medium. This is addressed in Supplemental Figure. 3b.

10. Later, using the resorufin assay, the authors demonstrate that "eructophagy" is dependent upon Beclin 1. This result is critical to the novelty of the whole story. However, later, when they set up experiments in later figures (Fig 9, 10), they do not use the Beclin 1KD as a control. This greatly weakens those figures, because it is impossible in the later figures to be sure that eructophagy is the key mechanism behind the transfer of signals from one macrophage to the other.

This has been addressed in Figure. 9 (d and e). Additional data has been incorporated to show DNA release in Beclin 1 deficient macrophages. Furthermore, data showing the effect of Beclin1 deficiency in FMLF release during eructophagy has been incorporated into Figure 10 (g and h).

11. Throughout the manuscript, there is a low standard for data quality, succinct communication, and transparency with adherence to conventional standards.

The manuscript has been edited and formatted as per Nature Communication's guidelines and each point that the reviewer raised has been addressed.

Specific examples include:

- none of the movies is annotated with arrows and generally fail to communicate the results the authors intended, as the reviewer lacks capacity to orient without processing and annotation.

This has been addressed. Arrows have been added to each video.

- there are several parts of the manuscript that have redundant sections. The introduction is conceptually engaging but could and should be greatly reduced in length.

We have edited the text of the manuscript, including the introduction to be more concise.

- Surprisingly, unless I missed it, I could not find a listing of author contributions/roles. No grant funding or other financial sourcing was listed. Some authors are listed with past affiliations. It may be appropriate to list Herbert Virgin, for instance, as being affiliated with his former university of employment if that is where he was when he contributed to the work in whatever role he played. However, it would also be important in the interest of transparency and convention to list his present institution in the acknowledgements. Indeed, the Acknowledgements section was, in contrast to other sections of the manuscript, inappropriately brief.

This information has all been added to the revised manuscript.

REVIEWER COMMENTS

Reviewer #1 (Remarks to the Author):

The authors have done a good job addressing most of my concerns.

Reviewer #2 (Remarks to the Author):

The manuscript by Greene and colleagues has been revised and addresses all the concerns raised in the previous review. I find the data compelling and the additional experiments fully convincing. I have one minor point that, while I accept fully that the process detailed here is novel and worthy of publication, the authors do overstate one aspect of the biology of PAMP accessibility in both the Introduction (line 70) and Discussion (line 447), regarding the sequestered nature of PAMPs released by intracellular pathogens. Mycobacterium-infected macrophages have been shown to release lipid PAMPs via exosomes, and these are highly immunomodulatory in nature.

Beatty, W.L., Rhoades, E.R., Ullrich, H.J., Chatterjee, D. and Russell, D.G. (2000) Trafficking and release of mycobacterial lipids from infected macrophages. *Traffic*. 1. 235-247.

Beatty, W., Ullrich, H-J. and Russell, D.G. (2001) Mycobacterial surface moieties are released from infected macrophages by a constitutive exocytic event. *Eur. J. Cell. Biol.* 80. 31-40.

Rhoades, E.R., Hsu, F-F., Torrelles, J.B., Chatterjee, D. and Russell, D.G. (2003) Identification and macrophage-activating activity of glycolipids released from intracellular Mycobacterium spp. *Mol. Microbiol.* 48. 875-888.

Response to reviewers

Reviewer #1 (Remarks to the Author):

The authors have done a good job addressing most of my concerns.

We thank the reviewer for their comments.

Reviewer #2 (Remarks to the Author):

The manuscript by Greene and colleagues has been revised and addresses all the concerns raised in the previous review. I find the data compelling and the additional experiments fully convincing. I have one minor point that, while I accept fully that the process detailed here is novel and worthy of publication, the authors do overstate one aspect of the biology of PAMP accessibility in both the Introduction (line 70) and Discussion (line 447), regarding the sequestered nature of PAMPs released by intracellular pathogens. Mycobacterium-infected macrophages have been shown to release lipid PAMPs via exosomes, and these are highly immunomodulatory in nature.

Beatty, W.L., Rhoades, E.R., Ullrich, H.J., Chatterjee, D. and Russell, D.G. (2000) Trafficking and release of mycobacterial lipids from infected macrophages. *Traffic*. 1. 235-247.

Beatty, W., Ullrich, H-J. and Russell, D.G. (2001) Mycobacterial surface moieties are released from infected macrophages by a constitutive exocytic event. *Eur. J. Cell. Biol.* 80. 31-40.

Rhoades, E.R., Hsu, F-F., Torrelles, J.B., Chatterjee, D. and Russell, D.G. (2003) Identification and macrophage-activating activity of glycolipids released from intracellular Mycobacterium spp. *Mol. Microbiol.* 48. 875-888.

We thank the reviewer for their comments, and we hope that the reviewer accepts the amended text and addition of the references to the manuscript.